# FEATURE-ALIGNED N-BEATS WITH SINKHORN DIVERGENCE

**Joonhun Lee**[*], **Myeongho Jeon**[*] **& Myungjoo Kang**[†]
Seoul National University
{niceguy718,andyjeon,mkang}@snu.ac.kr

**Kyunghyun Park**[†]
Nanyang Technological University
kyunghyun.park@ntu.edu.sg

## ABSTRACT

We propose Feature-aligned N-BEATS as a domain-generalized time series forecasting model. It is a nontrivial extension of N-BEATS with doubly residual stacking principle (Oreshkin et al. [45]) into a representation learning framework. In particular, it revolves around marginal feature probability measures induced by the intricate composition of residual and feature extracting operators of N-BEATS in each stack and aligns them stack-wise via an approximate of an optimal transport distance referred to as the Sinkhorn divergence. The training loss consists of an empirical risk minimization from multiple source domains, i.e., forecasting loss, and an alignment loss calculated with the Sinkhorn divergence, which allows the model to learn invariant features stack-wise across multiple source data sequences while retaining N-BEATS's interpretable design and forecasting power. Comprehensive experimental evaluations with ablation studies are provided and the corresponding results demonstrate the proposed model's forecasting and generalization capabilities.

## 1 INTRODUCTION

Machine learning models typically presume that the loss minimization from training data results in reasonable performance on a target environment, i.e., empirical risk minimization [56]. However, when using such models in the real world, the target environment is likely to deviate from the training data, which poses a significant challenge for a well-adaptive model to the target environment. This is related to the concept of *domain shift* [49].

A substantial body of research has been dedicated to developing frameworks that can accommodate the domain shift issue [6, 7, 20]. In particular, classification tasks have been the predominant focus [30, 32, 58, 59, 66]. As an integral way for modeling sequential data in broad domains such as finance, operation research, climate modeling, and biostatistics, *time series forecasting* has been a big part of machine learning fields. Nevertheless, the potential domain shift issue for common forecasting tasks has not been considered intensively compared to classification tasks, but only a few articles addressing this can be named [25, 26].

The goal of this article is to propose a resolution for the domain shift issue within time series forecasting tasks, namely a *domain-generalized* time series forecasting model. In particular, the proposed model is built upon a deep learning model which is N-BEATS [45, 46], and a representation learning toolkit which is the *feature alignment*. N-BEATS revolves around a doubly residual stacking principle and enhances the forecasting capabilities of multilayer perceptron (MLP) architectures without resorting to any traditional machine learning methods. On the other hand, it is well-known that aligning marginal feature measures enables machine learning models to capture invariant features across distinctive domains [8]. Indeed, in the context of classification tasks, many references [30, 38, 42, 65] demonstrated that the feature alignment mitigates the domain shift issue.

It is important to highlight that the model is not a straightforward combination of the established components but a nontrivial extension that poses several challenges. First, N-BEATS does not allow the feature alignment in a 'one-shot' unlike the aforementioned references. This is because it is a hierarchical multi-stacking architecture in which each stack consists of several blocks and is connected to each other by residual operations and feature extractions. In response to this, we devise the

---

[*]Equal contribution    [†]Co-corresponding authors

*stack-wise* alignment that is a minimization of divergences between marginal feature measures on a stack-wise basis. The stack-wise alignment enables the model to learn feature invariance with an ideal frequency of propagation. Indeed, instead of aligning every block for each stack, single alignment for each stack mitigates gradient vanishing/exploding issue [47] via sparsely propagating loss while preserving the interpretability of N-BEATS and ample semantic coverage [45, Section 3.3].

Second, the stack-wise alignment demands an efficient and accurate method for measuring divergence between measures. Indeed, the alignment is inspired by the error analysis of general domain generalization models given in [1, Theorem 1], in which empirical risk minimization loss and pairwise $\mathcal{H}$-divergence loss between marginal feature measures are the trainable components among the total error terms without any target domain information. In particular, since the feature alignment requires the calculation of pairwise divergences for multiple stacks (due to the doubly residual stacking principle), the computational load steeply increases as either the number of source domains or that of stacks increases. On the other hand, from the perspective of accuracy and efficiency, the $\mathcal{H}$-divergence is notoriously challenging to be used in practice [6, 28, 31, 54].

For a suitable toolkit, we adopt the Sinkhorn divergence which is an efficient approximation for the classic optimal transport distances [17, 21, 50]. This choice is motivated by the substantial theoretical evidences of optimal transport distances. Indeed, in the adversarial framework, optimal transport distances have been essential for theoretical evidences and calculation of divergences between pushforward measures induced by a generator and a target measure [21, 53, 62, 66]. In particular, the computational efficiency of the Sinkhorn divergence and fluent theoretical results by [13, 14, 17, 21] are crucial for our choice among other optimal transport distances. Thereby, the training objective is to minimize the empirical risk and the stack-wise Sinkhorn divergences (Section 3.3).

**Contributions.** To provide an informative procedure of stack-wise feature alignment, we introduce a concrete mathematical formulation of N-BEATS (Section 2), which enables to define the pushforward feature measures induced by the intricate residual operations and the feature extractions for each stack (Section 3.1). From this, we make use of theoretical properties of optimal transport problems to show a representation learning bound for the stack-wise feature alignment with the Sinkhorn divergence (Theorem 3.6), which justifies the feasibility of Feature-aligned N-BEATS. To embrace comprehensive domain generalization scenarios, we use real-world data and evaluate the proposed method under three distinct protocols based on the domain shift degree. We show that the model consistently outperforms other forecasting models. In particular, our method exhibits outstanding generalization capabilities under severe domain shift cases (Table 1). We further conduct ablation studies to support the choice of the Sinkhorn divergence in our model (Table 2).

**Related literature.** For time series forecasting, deep learning architectures including recurrent neural networks [4, 9, 24, 51, 52] and convolutional neural networks [11, 34] have achieved significant progress. Recently, a prominent shift has been observed towards transformer architectures leveraging self-attention mechanisms [33, 35, 60, 61, 67, 68]. Despite their innovations, concerns have been raised regarding the inherent permutation invariance in self-attention, which potentially leads to the loss of temporal information [63]. On the other hand, [10, 45] empirically show that MLP-based architectures would mitigate such a disadvantage and even surpass the transformer-based models.

Regarding the domain shifts for time series modeling, [25] proposed a technique that selects samples from source domains resembling the target domain, and employs regularization to encourage learning domain invariance. [26] designed a shared attention module paired with a domain discriminator to capture domain invariance. [46] explored domain generalization from a meta-learning perspective without the information on the target domain. Nonetheless, an explicit toolkit and concrete formulation for domain generalization are not considered therein.

The remainder of the article is organized as follows. In Section 2, we set the domain generalization problem in the context of time series forecasting, review the doubly residual stacking architecture of N-BEATS, and introduce the error analysis for domain generalization models. Section 3 is devoted to defining the marginal feature measures inspiring the stack-wise alignment, introducing the Sinkhorn divergence together with the corresponding representation learning bound, and presenting the training objective with the corresponding algorithm. In particular, Figure 1 therein illustrates the overall architecture of Feature-aligned N-BEATS. In Section 4, comprehensive experimental evaluations are provided. Section 5 concludes the paper. Other technical descriptions, visualized results, and ablation studies are given in Appendix.

## 2 BACKGROUND

**Notations.** Let $\mathcal{X} := \mathbb{R}^\alpha$ and $\mathcal{Y} := \mathbb{R}^\beta$ be the input and output spaces, respectively, where $\alpha$ and $\beta$ denote the lookback and forecast horizons, respectively. Let $\mathcal{Z} := \mathbb{R}^\gamma$ be the latent space with $\gamma$ representing the feature dimension. We further denote by $\widetilde{\mathcal{Z}} \subset \mathcal{Z}$ a subspace of $\mathcal{Z}$. All the aforementioned spaces are equipped with the Euclidean norm $\|\cdot\|$. Define by $\mathcal{P} := \mathcal{P}(\mathcal{X} \times \mathcal{Y})$ the set of all Borel joint probability measures on $\mathcal{X} \times \mathcal{Y}$. For any $\mathbb{P} \in \mathcal{P}$, denotes by $\mathbb{P}_\mathcal{X}$ and $\mathbb{P}_\mathcal{Y}$ corresponding marginal probability measures on $\mathcal{X}$ and $\mathcal{Y}$, respectively. We further define by $\mathcal{P}(\mathcal{X})$ and $\mathcal{P}(\widetilde{\mathcal{Z}})$ the sets of all Borel probability measures on $\mathcal{X}$ and $\widetilde{\mathcal{Z}}$, respectively.

**Domain generalization in time series forecasting.** There are multiple source domains $\{\mathcal{D}^k\}_{k=1}^K$ with $K \geq 2$ and target (unseen) domain $\mathcal{D}^T$. Assume that each $\mathcal{D}^k$ is equipped with $\mathbb{P}^k \in \mathcal{P}$ and the same holds for $\mathcal{D}^T$ with $\mathbb{P}^T \in \mathcal{P}$ and that sequential data for each domain are sampled from corresponding joint distribution. Let $l : \mathcal{Y} \times \mathcal{Y} \to \mathbb{R}_+$ be a loss function. Then, the objective is to derive a prediction model $\mathfrak{F} : \mathcal{X} \to \mathcal{Y}$ such that $\mathfrak{F}(\mathbf{s}_{t-\alpha+1}, \cdots, \mathbf{s}_t) \approx \mathbf{s}_{t+1}, \cdots \mathbf{s}_{t+\beta}$ for $\mathbf{s} = (\mathbf{s}_{t-\alpha+1}, \cdots, \mathbf{s}_t) \times (\mathbf{s}_{t+1}, \cdots \mathbf{s}_{t+\beta}) \sim \mathbb{P}^T$, by leveraging on $\{\mathbb{P}^k\}_{k=1}^K$, i.e.,

$$\inf_\mathfrak{F} \ \mathcal{L}(\mathfrak{F}), \quad \text{with} \quad \mathcal{L}(\mathfrak{F}) := \frac{1}{K}\sum_{k=1}^K \mathbb{E}_{(x,y)\sim\mathbb{P}^k}\left[l\big(\mathfrak{F}(x), y\big)\right]. \tag{2.1}$$

**Doubly residual stacking architecture.** The main architecture of N-BEATS equipped with the doubly residual stacking principle from [10, 45] is summarized as follows: for $M, L \in \mathbb{N}$, the model comprises $M$ stacks, with each stack consisting of $L$ blocks. The blocks share the same model weight within each respective stack and are recurrently operated based on the double residual stacking principle. More precisely, an $m$-th stack derives the principle in a way that for $x_{m,1} \in \mathcal{X}$,

$$\hat{y}_m := \sum_{l=1}^L (\xi_\downarrow^m \circ \psi^m)(x_{m,l}), \ \ x_{m,l} := x_{m,l-1} - (\xi_\uparrow^m \circ \psi^m)(x_{m,l-1}), \quad l = 2, \ldots, L, \tag{2.2}$$

where $\psi^m : \mathcal{X} \to \mathcal{Z}$ extracts features $\psi^m(x_{m,l}) \in \mathcal{Z}$ from the inputs $x_{m,l} \in \mathcal{X}$ for each layer $l$, and $(\xi_\downarrow^m, \xi_\uparrow^m) : \mathcal{Z} \to \mathcal{Y} \times \mathcal{X}$ generates both forecasts $(\xi_\downarrow^m \circ \psi^m)(x_{m,l}) \in \mathcal{Y}$ and backcasts $(\xi_\uparrow^m \circ \psi^m)(x_{m,l}) \in \mathcal{X}$ branches. Note that $\hat{y}_m \in \mathcal{Y}$ represents the $m$-th forecast obtained through the hierarchical aggregation of each block's forecast, and that the last backcast $x_{m,L} \in \mathcal{X}$, derived by a residual sequence from blocks, serves as an input for the next stack, except for the case $m = M$.

Once the hierarchical aggregation of all stacks and the residual operations are completed, the model $\mathfrak{F}$ for the doubly residual stacking architecture is given as follows: for $(x, y) \sim \mathbb{P}^T$ and $x_{1,1} := x$,

$$y \approx \mathfrak{F}(x; \Psi, \Xi_\downarrow, \Xi_\uparrow) := \sum_{m=1}^M \hat{y}_m, \quad x_{m,1} := x_{m-1,L}, \quad m = 2, \ldots, M, \tag{2.3}$$

subject to $\hat{y}_m$ and $x_{m-1,L}$ given in (2.2), where

$$\Psi := \{\psi^m\}_{m=1}^M, \ \ \Xi_\downarrow := \{\xi_\downarrow^m\}_{m=1}^M, \ \ \Xi_\uparrow := \{\xi_\uparrow^m\}_{m=1}^M, \tag{2.4}$$

are implemented by fully connected layers. For further details, refer to Appendix A.

**Domain-invariant feature representation.** After the investigation on the error analysis for domain adaptation models by [64], an extended version for domain generalization models is provided by [1]. This provides us an insight for developing a domain generalization toolkit within the context of doubly residual stacking models.

In the following, we restate Theorem 1 in [1]. To that end, we introduce some notations. Let $\mathcal{H}$ be the set of hypothesis functions $h : \mathcal{X} \to [0, 1]$ and let $\widetilde{\mathcal{H}} := \{\text{sgn}(|h(\cdot) - h'(\cdot)| - t) : h, h' \in \mathcal{H}, \ t \in [0, 1]\}$. The $\mathcal{H}$-divergence is defined by $d_\mathcal{H}(\mathbb{P}'_\mathcal{X}, \mathbb{P}''_\mathcal{X}) := 2\sup_{h\in\mathcal{H}} |\mathbb{E}_{x\sim\mathbb{P}'_\mathcal{X}}[\mathbf{1}_{\{h(x)=1\}}] - \mathbb{E}_{x\sim\mathbb{P}''_\mathcal{X}}[\mathbf{1}_{\{h(x)=1\}}]|$ for any $\mathbb{P}'_\mathcal{X}, \mathbb{P}''_\mathcal{X} \in \mathcal{P}(\mathcal{X})$. The $\widetilde{\mathcal{H}}$-divergence $d_{\widetilde{\mathcal{H}}}(\cdot, \cdot)$ is defined analogously, with $\mathcal{H}$ replaced by $\widetilde{\mathcal{H}}$. Furthermore, denote by $R^k(\cdot) : \mathcal{H} \to \mathbb{R}$ and $R^T(\cdot) : \mathcal{H} \to \mathbb{R}$ the expected risk under the source measures $\mathbb{P}^k$, $k = 1, \ldots, K$, and the target measure $\mathbb{P}^T$, respectively.

**Proposition 2.1.** *Let $\Delta_K$ be a (K-1)-dimensional simplex such that each component $\pi$ represents a convex weight. Set $\Lambda := \{\sum_{k=1}^{K} \pi_i \mathbb{P}_{\mathcal{X}}^k | \pi \in \Delta_K\}$ and let $\mathbb{P}^* := \sum_{k=1}^{K} \pi_k^* \mathbb{P}_{\mathcal{X}}^k \in \arg\min_{\mathbb{P}'_{\mathcal{X}} \in \Lambda} d_{\mathcal{H}}(\mathbb{P}_{\mathcal{X}}^T, \mathbb{P}'_{\mathcal{X}})$. Then, the following holds: for any $h \in \mathcal{H}$,*

$$R^T(h) \leq \Sigma_{k=1}^{K} \pi_k^* R^k(h) + d_{\mathcal{H}}(\mathbb{P}_{\mathcal{X}}^T, \mathbb{P}_{\mathcal{X}}^*) + \max_{i,j \in \{1,\dots,K\},\ i \neq j} d_{\widetilde{\mathcal{H}}}(\mathbb{P}_{\mathcal{X}}^i, \mathbb{P}_{\mathcal{X}}^j) + \lambda_{(\mathbb{P}_{\mathcal{X}}^T, \mathbb{P}_{\mathcal{X}}^*)},$$

*with $\lambda_{(\mathbb{P}_{\mathcal{X}}^T, \mathbb{P}_{\mathcal{X}}^*)} := \min\{\mathbb{E}_{x \sim \mathbb{P}_{\mathcal{X}}^T}[|\sum_{k=1}^{K} \pi_k^* f^k(x) - f^T(x)|], \mathbb{E}_{x \sim \mathbb{P}_{\mathcal{X}}^*}[|\sum_{k=1}^{K} \pi_k^* f^k(x) - f^T(x)|]\}$, where $f^k$, $k = 1, \dots, K$, denotes a true labeling function under $\mathbb{P}^k$, i.e., $y = f^k(x)$ for $(x, y) \sim \mathbb{P}^k$, and similarly $f^T$ denotes a true labeling function under $\mathbb{P}^T$.*

While the upper bound of $R^T(\cdot)$ consists of four terms, only the first and third terms (representing the source risks $\{R^k(\cdot)\}_{k=1}^{K}$ and the pairwise divergences $\{d_{\widetilde{\mathcal{H}}}(\mathbb{P}_{\mathcal{X}}^i, \mathbb{P}_{\mathcal{X}}^j)\}_{i \neq j}^{K}$ across all marginal feature measures, respectively) are learnable without the target domain information.

## 3 METHOD

### 3.1 MARGINAL FEATURE MEASURES

Aligning marginal feature measures is a predominant approach in domain-invariant representation learning [20, 55]. In particular, the marginal feature measures $\{g_{\#}\mathbb{P}_{\mathcal{X}}^k\}_{k=1}^{K}$ are defined as push-forward measures induced by a given feature map $g : \mathcal{X} \to \mathcal{Z}$ from $\{\mathbb{P}_{\mathcal{X}}^k\}_{k=1}^{K}$, i.e., $g_{\#}\mathbb{P}_{\mathcal{X}}^k(E) = \mathbb{P}_{\mathcal{X}}^k \circ g^{-1}(E)$ for any Borel set $E$ in $\mathcal{Z}$.

However, defining such measures for doubly residual architectures poses some challenges. Indeed, as discussed in Section 2, N-BEATS includes multiple feature extractors $\Psi = \{\psi^m\}_{m=1}^{M}$ as defined in (2.2), where each extractor $\psi^m$ takes a sampled input passing through multiple residual operations of previous stacks and the input is recurrently processed within each stack by the residual operations $\Xi_{\downarrow}$ and $\Xi_{\uparrow}$. The scaling factor represents domain-specific characteristics that exhibit noticeable variations. This can lead to an excessive focus on scale adjustments in the aligning process, potentially neglecting crucial features, such as seasonality or trend.

To resolve these difficulties, we propose a stack-wise alignment of feature measures on subspace $\widetilde{\mathcal{Z}} \subseteq \mathcal{Z}$. This involves defining measures for each stack through the compositions of feature extractions in $\Psi = \{\psi^m\}_{m=1}^{M}$, backcasting operators in $\Xi_{\uparrow} = \{\xi_{\uparrow}^m\}_{m=1}^{M}$ given in (2.2), and a normalization function.

**Definition 3.1.** Let $\sigma : \mathcal{Z} \to \widetilde{\mathcal{Z}}$ be a normalizing function satisfying $C_{\sigma}$-Lipschitz continuity, i.e., $\|\sigma(z) - \sigma(z')\| \leq C_{\sigma}\|z - z'\|$ for $z, z' \in \mathcal{Z}$. Given $\psi^m : \mathcal{X} \to \mathcal{Z}$ defined in (2.2), the operators $r^m : \mathcal{X} \to \mathcal{X}$ and $g^m : \mathcal{X} \to \mathcal{Z}$ are defined as:

$$r^m(x) := x - (\xi_{\uparrow}^m \circ \psi^m)(x), \tag{3.1}$$

$$g^m(x) := (\psi^m \circ (r^m)^{(L-1)} \circ (r^{m-1})^{(L)} \circ \cdots \circ (r^1)^{(L)})(x), \tag{3.2}$$

where $(r^m)^{(L)}$ denotes $L$-times composition of $r^m$, with $(r^m)^{(L-1)}(x) := x$ for $L - 1 = 0$ and $g^m = (\psi^m \circ (r^m)^{(L-1)})$ for $m = 1$. Then the set of marginal feature measures in the $m$-th stack, $m = 1, \cdots, M$, is defined by

$$\{(\sigma \circ g^m)_{\#}\mathbb{P}_{\mathcal{X}}^k\}_{k=1}^{K},$$

where each $(\sigma \circ g^m)_{\#}\mathbb{P}_{\mathcal{X}}^k$ is a pushforward of $\mathbb{P}_{\mathcal{X}}^k \in \{\mathbb{P}_{\mathcal{X}}^k\}_{k=1}^{K}$ induced by $\sigma \circ g^m : \mathcal{X} \to \widetilde{\mathcal{Z}}$.

**Remark 3.2.** The normalization function $\sigma$ helps the model to learn invariant features by mitigating the influence of the scale information of each domain. Furthermore, the Lipschitz condition on $\sigma$ prevents gradient explosion during model updates. There are two representatives for $\sigma$: (i) softmax : $\mathcal{Z} \to \widetilde{\mathcal{Z}} = (0, 1)^{\gamma}$ where $\text{softmax}(z)_j = e^{z_j}/\sum_{i=1}^{\gamma} e^{z_i}, j = 1, \dots, \gamma$; (ii) tanh: $\mathcal{Z} \to \widetilde{\mathcal{Z}} = (-1, 1)^{\gamma}$ where $\tanh(z)_j = (e^{2z_j} - 1)/(e^{2z_j} + 1), j = 1, \dots, \gamma$. Both are 1-Lipschitz continuous, i.e., $C_{\sigma} = 1$. In Appendix G (see Table 9), we provide the ablation study under these functions, in addition to the case without the normalization.

**Remark 3.3.** Embedding feature alignment 'block-wise' for every stack results in recurrent operations within each stack and redundant gradient flows. This redundancy can cause exploding or

vanishing gradients for long-term forecasting [47]. Our stack-wise feature alignment addresses these problems by sparsely propagating the loss. It also maintains ample alignment coverage related to semantics since the stack serves as a semantic extraction unit in [45]. Further heuristic demonstration is provided in Appendix G.1.

The operator $g^m$ in (3.2) accumulates features up to the $m$-th stack accounting for the previous $m-1$ residual operations. Despite the complex composition of $\Psi$ and $\Xi_\uparrow$, the fully connected layers in them exhibit Lipschitz continuity [57, Section 6], which ensures the Lipschitz continuity of $g^m$. From this observation and Remark 3.2, we state the lemma below, with its proof in Appendix B:

**Lemma 3.4.** *Let $C_\sigma > 0$ be given in Definition 3.1. Denote for $m = 1, \cdots, M$ by $C_m > 0$ and $C_{m,\uparrow} > 0$ the Lipschitz constants of $\psi^m$ and $\xi_\uparrow^m$, respectively. Then $(\sigma \circ g^m)$ is $C_{\sigma \circ g^m}$-Lipschitz continuous with*

$$C_{\sigma \circ g^m} = C_\sigma C_m (1 + C_m C_{m,\uparrow})^{L-1} \Pi_{n=1}^{m-1} (1 + C_n C_{n,\uparrow})^L, \quad for \ \ m = 2, \cdots, M,$$

*and $C_{\sigma \circ g^m} = C_\sigma C_m (1 + C_m C_{m,\uparrow})^{L-1}$ for $m = 1$.*

By the doubly residual principle, $\{g^m\}_{m=1}^M$ are inseparable for $\Psi$ and $\Xi_\uparrow$. However, the stack-wise alignment via regularizing $\{g^m\}_{m=1}^M$ potentially deteriorates the backcasting power of $\Xi_\uparrow$, which could lead to performance degradation of the model. Instead, we conduct the alignment by regularizing exclusively on feature extractors $\Psi$. More precisely, this alignment of marginal feature measures from Definition 3.1 is defined as follows: given $\Xi_\uparrow = \{\xi_\uparrow^m\}_{m=1}^M$,

$$\inf_\Psi \left\{ \sum_{m=1}^M \max_{i,j \in \{1,\cdots,K\}, \ i \neq j} d\left( (\sigma \circ g^m)_\# \mathbb{P}_\mathcal{X}^i, (\sigma \circ g^m)_\# \mathbb{P}_\mathcal{X}^j \right) \right\}, \tag{3.3}$$

where $d(\cdot, \cdot) : \mathcal{P}(\widetilde{\mathcal{Z}}) \times \mathcal{P}(\widetilde{\mathcal{Z}}) \to \mathbb{R}_+$ is a divergence or distance between given measures. The illustration of the stack-wise alignment is provided in Figure 3 (in Appendix A).

Note that the third term in Proposition 2.1, i.e., $\max_{i,j \in \{1,\dots,K\}, \ i \neq j} d_{\widetilde{\mathcal{H}}}(\mathbb{P}_\mathcal{X}^i, \mathbb{P}_\mathcal{X}^j)$, and the stack-wise alignment in (3.3) are perfectly matched once $d(\cdot, \cdot)$ is specified as the $\mathcal{H}$-divergence. However, the empirical estimation for the $\mathcal{H}$-divergence is notoriously difficult [6, 7, 32, 54]. These concerns become even more pronounced in the proposed method due to the stack-wise alignment necessitating $MK(K-1)/2$-times calculation of pairwise divergence, implying heavy computational load. Meanwhile, a substantial body of literature regarding the domain invariant feature learning adopts other alternatives for the $\mathcal{H}$-divergence, and among them [28, 29, 53, 66], optimal transport distances have been dominant due to their in-depth theoretical ground. In line with this, in the following section, we introduce an optimal transport distance as a relevant choice for $d(\cdot, \cdot)$.

## 3.2 SINKHORN DIVERGENCE ON MEASURES

In the adversarial framework [21, 53, 62, 66], optimal transport distances have been adopted for training generators to make corresponding pushforward measures close to a given target measure. In particular, the Sinkhorn divergence, an approximate of an entropic regularized optimal transport distance, is shown to be an efficient method to address intensive calculations of divergence between empirical measures [13, 17, 21]. As the stack-wise alignment given in (3.3) leverages on a number of calculations of divergences and hence requires an efficient and accurate toolkit for feasible training, we adopt the Sinkhorn divergence as the relevant one for $d(\cdot, \cdot)$.

To define the Sinkhorn divergence, let us introduce the regularized quadratic Wasserstein-2 distance. To that end, let $\epsilon$ be the entropic regularization degree and $\Pi(\mu, \nu; \widetilde{\mathcal{Z}})$ is the space of all couplings, i.e., transportation plans, the marginals of which are respectively $\mu, \nu \in \mathcal{P}(\widetilde{\mathcal{Z}})$. Then the regularized quadratic Wasserstein-2 distance defined on $\widetilde{\mathcal{Z}}$ is defined as follows: for $\epsilon \geq 0$,

$$\mathcal{W}_{\epsilon, \widetilde{\mathcal{Z}}}(\mu, \nu) := \inf_{\pi \in \Pi(\mu,\nu;\widetilde{\mathcal{Z}})} \left\{ \int_{\widetilde{\mathcal{Z}} \times \widetilde{\mathcal{Z}}} \left( \|x - y\|^2 + \epsilon \log \left( \frac{d\pi(x,y)}{d\mu(x)d\nu(y)} \right) \right) d\pi(x,y) \right\}. \tag{3.4}$$

By replacing $\widetilde{\mathcal{Z}}$ with $\mathcal{X}$, one can define by $\mathcal{W}_{\epsilon, \mathcal{X}}(\cdot, \cdot)$ the corresponding regularized distance on $\mathcal{X}$.

The entropic term attached with $\epsilon$ in (3.4) is known to improve computational stability of the Wasserstein-2 distance, whereas it causes a bias on corresponding estimator. To alleviate this, according to [12], we adopt the following debiased version of the regularized distance:

**Definition 3.5.** For $\epsilon \geq 0$, the Sinkhorn divergence is

$$\widehat{\mathcal{W}}_{\epsilon,\widetilde{\mathcal{Z}}}(\mu,\nu) := \mathcal{W}_{\epsilon,\widetilde{\mathcal{Z}}}(\mu,\nu) - \frac{1}{2}\left(\mathcal{W}_{\epsilon,\widetilde{\mathcal{Z}}}(\nu,\nu) + \mathcal{W}_{\epsilon,\widetilde{\mathcal{Z}}}(\mu,\mu)\right), \quad \mu,\nu \in \mathcal{P}(\widetilde{\mathcal{Z}}). \qquad (3.5)$$

Using the duality of the regularized optimal transport distance from [48, Remark 4.18 in Section 4.4] and the Lipschitz continuity of $\{\sigma \circ g^m\}_{m=1}^M$ from Lemma 3.4, we present the following theorem, substantiating the well-definedness and feasibility of our stack-wise alignment via $\widehat{\mathcal{W}}_{\epsilon,\widetilde{\mathcal{Z}}}(\cdot,\cdot)$. The proof is provided in Appendix B.

**Theorem 3.6.** Let $C_{\sigma \circ g^m} > 0$ be as in Lemma 3.4 and define $C := \sum_{m=1}^M \max\{(C_{\sigma \circ g^m})^2, 1\}$. Then the following holds: for $\epsilon \geq 0$,

$$\sum_{m=1}^M \max_{i,j \in \{1,\cdots,K\},\ i \neq j} \widehat{\mathcal{W}}_{\epsilon,\widetilde{\mathcal{Z}}}\left((\sigma \circ g^m)_\# \mathbb{P}_\mathcal{X}^i, (\sigma \circ g^m)_\# \mathbb{P}_\mathcal{X}^j\right) \leq C \max_{i,j \in \{1,\cdots,K\},\ i \neq j} \mathcal{W}_{\epsilon,\mathcal{X}}\left(\mathbb{P}_\mathcal{X}^i, \mathbb{P}_\mathcal{X}^j\right).$$

In [44, Lemma 3 & Proposition 6], representation learning bounds under the maximum mean discrepancy and the regularized distance in (3.4) are investigated for a single-layered fully connected network. With similar motivation, Theorem 3.6 represents a learning bound for the stack-wise alignment loss under the Sinkhorn divergence as the entropic regularized distance between source domains' measures. While the Lipschitz continuity of $\{\sigma \circ g^m\}_{m=1}^M$ allows a nice bound, there exists room for having a tighter bound by deriving the smallest Lipschitz constant [57] and applying the spectral normalization [40], which will be left for the future extension. Further discussions on the choice of the Sinkhorn divergence and on Theorem 3.6 are provided in Appendix C.

### 3.3 TRAINING OBJECTIVE AND ALGORITHM

From Sections 3.1 and 3.2, we define the training objective and corresponding algorithm. To that end, denote by $\Phi := \{\phi_m\}_{m=1}^M$, $\Theta_\downarrow := \{\theta_{m,\downarrow}\}_{m=1}^M$, and $\Theta_\uparrow := \{\theta_{m,\uparrow}\}_{m=1}^M$ the parameters sets of the fully connected neural networks in the residual operators in $\Psi$, $\Xi_\downarrow$, and $\Xi_\uparrow$ given in (2.4). Then corresponding parameterized forms of the operators are given by

$$\Psi(\Phi) = \{\psi^m(\cdot;\phi_m)\}_{m=1}^M, \quad \Xi_\downarrow(\Theta_\downarrow) = \{\xi_\downarrow^m(\cdot;\theta_{m,\downarrow})\}_{m=1}^M, \quad \Xi_\uparrow(\Theta_\uparrow) = \{\xi_\uparrow^m(\cdot;\theta_{m,\uparrow})\}_{m=1}^M.$$

Then denote by $g_{\Phi,\Theta_\uparrow}^m := g^m(\cdot;\{\phi_n\}_{n=1}^m,\{\theta_{n,\uparrow}\}_{n=1}^m)$, $m = 1,\ldots,M$, the parameterized version of $g^m$ given in (3.2). Let $\mathcal{L}(\mathfrak{F}(\cdot,\cdot,\cdot))$ be the parameterized form of the forecasting loss given in (2.1) and $\mathcal{L}_{\text{align}}(\cdot,\cdot)$ be that of the alignment loss given in (3.3) under the Sinkhorn divergence, i.e.,

$$\mathcal{L}_{\text{align}}(\Phi,\Theta_\uparrow) := \sum_{m=1}^M \max_{i,j \in \{1,\ldots,K\},\ i \neq j} \widehat{\mathcal{W}}_{\epsilon,\widetilde{\mathcal{Z}}}\left((\sigma \circ g_{\Phi,\Theta_\uparrow}^m)_\# \mathbb{P}_\mathcal{X}^i, (\sigma \circ g_{\Phi,\Theta_\uparrow}^m)_\# \mathbb{P}_\mathcal{X}^j\right). \qquad (3.6)$$

We then provide the following training objective

$$\mathbf{L}_\lambda(\Phi,\Theta_\downarrow,\Theta_\uparrow) := \mathcal{L}(\mathfrak{F}(\Phi,\Theta_\downarrow,\Theta_\uparrow)) + \lambda \mathcal{L}_{\text{align}}(\Phi,\Theta_\uparrow). \qquad (3.7)$$

To update $(\Phi,\Theta_\downarrow,\Theta_\uparrow)$ according to (3.7), we calculate $m$-th stack divergence $\widehat{\mathcal{W}}_{\epsilon,\widetilde{\mathcal{Z}}}((\sigma \circ g_{\Phi,\Theta_\uparrow}^m)_\# \mathbb{P}_\mathcal{X}^i, (\sigma \circ g_{\Phi,\Theta_\uparrow}^m)_\# \mathbb{P}_\mathcal{X}^j)$ as its empirical counterpart $\widehat{\mathcal{W}}_{\epsilon,\widetilde{\mathcal{Z}}}(\mu_{\Phi,\Theta_\uparrow}^{m,(i)}, \mu_{\Phi,\Theta_\uparrow}^{m,(j)})$, where the corresponding empirical measures $\{\mu_{\Phi,\Theta_\uparrow}^{m,(k)}\}_{k=1}^K$ are given as follow: for $k = 1,\cdots,K$,

$$\mu_{\Phi,\Theta_\uparrow}^{m,(k)} := \frac{1}{B}\sum_{b=1}^B \delta_{\widetilde{z}_b^{(k)}}, \quad \text{with } \widetilde{z}_b^{(k)} := \sigma \circ g_{\Phi,\Theta_\uparrow}^m(x_b^{(k)}),$$

where $\{(x_b^{(k)}, y_b^{(k)})\}_{b=1}^B$ are sampled from $\mathcal{D}^k$, and $B$ and $\delta_z$ denote a mini-batch size and the Dirac measure centered on $z \in \widetilde{\mathcal{Z}}$, respectively.

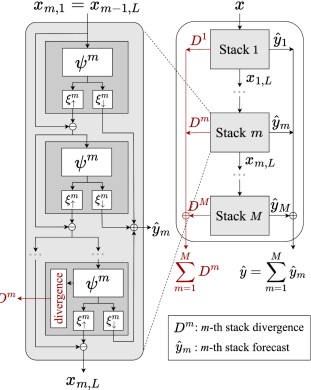

Figure 1: Illustration of Feature-aligned N-BEATS.

As mentioned in Section 3.1, the alignment loss $\mathcal{L}_{\text{align}}(\Phi, \Theta_\uparrow)$ is minimized by updating $\Phi$ for given $\Theta_\uparrow$, while $\{g^m_{\Phi,\Theta_\uparrow}\}^m_{m=1}$ are inseparable for $\Phi$ and $\Theta_\uparrow$. At the same time, the forecasting loss $\mathcal{L}(\mathfrak{F}(\Phi, \Theta_\downarrow, \Theta_\uparrow))$ is minimized by updating $(\Phi, \Theta_\downarrow, \Theta_\uparrow)$. To bring them together, we adopt the following alternatively updating optimization inspired from [19, Section 3.1]:

$$\Theta_\downarrow^*, \Theta_\uparrow^* := \underset{\Theta_\downarrow, \Theta_\uparrow}{\arg\min}\, \mathcal{L}(\mathfrak{F}(\Phi^*, \Theta_\downarrow, \Theta_\uparrow)), \quad \Phi^* := \underset{\Phi}{\arg\min}\, \mathbf{L}_\lambda(\Phi, \Theta_\downarrow^*, \Theta_\uparrow^*). \tag{3.8}$$

The training procedure on (3.8) is summarized in Algorithm 1 and the overall model architecture is illustrated in Figure 1, where we highlight the stack-wise alignment process (with 'red' color) not appearing in the original N-BEATS (see Figure 1 in [45]).

---

**Algorithm 1:** Training Feature-aligned N-BEATS.

---

**Requires:** $\eta$ (learning rate), $B$ (mini-batch size); Initialize $\Phi, \Theta_\downarrow, \Theta_\uparrow$;

1 **while** *not converged* **do**

2      Sample $\{(x_b^{(k)}, y_b^{(k)})\}_{b=1}^B$ from $\mathcal{D}^k$ & Initialize $\{\hat{y}_b^{(k)}\}_{b=1}^B \leftarrow 0, \ k = 1, \ldots, K$;

3      **for** $m = 1$ **to** $M$ **do**

4          **for** $k = 1$ **to** $K$ **do**

5              Compute $\{g^m_{\Phi,\Theta_\uparrow}(x_b^{(k)})\}_{b=1}^B$; Update $\hat{y}_b^{(k)} \leftarrow \hat{y}_b^{(k)} + \xi^m_\downarrow(g^m_{\Phi,\Theta_\uparrow}(x_b^{(k)}); \theta_{m,\downarrow}), \ b = 1, \ldots, B$;

6          **end**

7      **end**

8      Compute $\{\mu^{m,(k)}_{\Phi,\Theta_\uparrow}\}_{m=1}^M, \ k = 1, \ldots, K$; Update $\Phi$ such that for $m = 1, \ldots, M$,

9      $\phi_m \leftarrow \phi_m + \eta \nabla_{\phi_m} \left( \lambda \sum\limits_{n=1}^M \max\limits_{i,j \in \{1, \cdots, K\}, \ i \neq j} \widehat{\mathcal{W}}_{\epsilon, \widetilde{\mathcal{Z}}} \left( \mu^{n,(i)}_{\Phi,\Theta_\uparrow}, \mu^{n,(j)}_{\Phi,\Theta_\uparrow} \right) \right)$;

10      Update $(\Phi, \Theta_\downarrow, \Theta_\uparrow)$ such that for $m = 1, \ldots, M$,

11      $(\phi_m, \theta_{m,\downarrow}, \theta_{m,\uparrow}) \leftarrow (\phi_m, \theta_{m,\downarrow}, \theta_{m,\uparrow}) + \eta \frac{1}{K \cdot B} \sum\limits_{k=1}^K \sum\limits_{b=1}^B \nabla_{(\phi_m, \theta_{m,\downarrow}, \theta_{m,\uparrow})} l(\hat{y}_b^{(k)}, y_b^{(k)})$;

12 **end**

---

## 4 EXPERIMENTS

**Evaluation details.** Our evaluation protocol lies on two principles: (i) real-world scenarios and (ii) examination of various domain shift environments between the source and target domains. For (i), we use financial data from the Federal Reserve Economic Data (FRED)[1] and weather data from the National Centers for Environmental Information (NCEI)[2]. For (ii), let us define a set of semantically similar domains as *superdomain* denoted by $\mathcal{A}_i$, e.g., $i =$ FRED, NCEI. We then categorize the domain shift scenarios into *out-domain generalization* (ODG), *cross-domain generalization* (CDG), and *in-domain generalization* (IDG) such that

· ODG: $\{\mathcal{D}^k\}_{k=1}^K \subseteq \mathcal{A}_i \xrightarrow{\text{Shift } (i \neq j)} \mathcal{D}^T \in \mathcal{A}_j$;

· CDG: $\{\mathcal{D}^k\}_{k=1}^{p-1} \subseteq \mathcal{A}_i, \{\mathcal{D}^k\}_{k=p}^K \subseteq \mathcal{A}_j \ (2 \leq p \leq K) \xrightarrow{\text{Shift } (i \neq j)} \mathcal{D}^T \in \mathcal{A}_i$ s.t. $\{\mathcal{D}^k\}_{k=1}^{p-1} \cap \mathcal{D}^T = \emptyset$;

· IDG: $\{\mathcal{D}^k\}_{k=1}^K \subseteq \mathcal{A}_i \xrightarrow{\text{Shift } (i=j)} \mathcal{D}^T \in \mathcal{A}_i$ s.t. $\{\mathcal{D}^k\}_{k=1}^K \cap \mathcal{D}^T = \emptyset$.

The domain shift from source to target becomes increasingly pronounced in the sequence of IDG, CDG, and ODG, making it even more challenging to generalize. For detailed data configuration and domain specifications, refer to Appendix D.

**Benchmarks.** We compare our proposed approach with deep learning-based models, including transformer (e.g., Informer [67], Autoformer [61]), MLP-based models (e.g., LTSF-Linear models [63] with NLinear and Dlinear) and N-BEATS based models (e.g., N-BEATS [45] and N-HiTS [10]). Note that since the aforementioned time series models addressing domain shift [25, 26] still requires target domain data (due to their 'domain-adapted' framework), we do not consider their models into our domain-generalized protocol.

**Experimental details.** We adopt the symmetric mean absolute percentage error (sMAPE) for $\mathcal{L}(\mathfrak{F}(\cdot, \cdot, \cdot))$ given in (3.7) and use the softmax function for $\sigma$ given in Definition 3.1. The Sinkhorn

---

[1] https://fred.stlouisfed.org    [2] https://ncei.noaa.gov

Table 1: Domain generalization performance. The performance across all combinations of each ODG, CDG, and IDG scenario is provided (as the average of scenarios for each FRED and NCEI). The detailed description for N-BEATS-G and N-BEATS-I is provided in Appendix A. The notation '+ FA' stands for feature alignment. Each evaluation is conducted three times, with different random seeds. Values over 10,000 are labeled as 'NA'. Runtime is measured for a single iteration.

| Methods | | N-HiTS | + FA (Ours) | N-BEATS-I | + FA (Ours) | N-BEATS-G | + FA (Ours) | NLinear | DLinear | Autoformer | Informer |
|---|---|---|---|---|---|---|---|---|---|---|---|
| | | ODG | | | | | | | | | |
| FRED | sMAPE | 0.148 | **0.134** | 0.232 | 0.214 | 0.172 | 0.150 | 0.176 | 0.307 | 0.570 | 1.214 |
| | MASE | 0.060 | **0.057** | 0.069 | 0.065 | 0.061 | 0.059 | 48.150 | 2,214.48 | NA | NA |
| NCEI | sMAPE | 0.723 | **0.713** | 0.814 | 0.724 | 0.722 | 0.718 | 1.112 | 1.302 | 1.293 | 1.630 |
| | MASE | 0.561 | **0.512** | 0.754 | 0.663 | 0.663 | 0.561 | 0.516 | 2.737 | 2.869 | 3.311 | 5.784 |
| | | CDG | | | | | | | | | |
| FRED | sMAPE | 0.124 | **0.123** | 0.181 | 0.179 | 0.139 | 0.133 | 0.176 | 0.536 | 0.893 | 1.143 |
| | MASE | 0.058 | **0.057** | 0.064 | 0.062 | 0.059 | 0.058 | 60.929 | 2,554.27 | NA | NA |
| NCEI | sMAPE | 0.742 | **0.718** | 0.731 | **0.718** | 0.763 | **0.718** | 1.096 | 1.086 | 1.273 | 1.437 |
| | MASE | 0.581 | **0.482** | 0.822 | 0.755 | 0.608 | 0.582 | 2.734 | 2.787 | 3.233 | 4.147 |
| | | IDG | | | | | | | | | |
| FRED | sMAPE | 0.119 | **0.115** | 0.137 | 0.136 | 0.143 | 0.119 | 0.197 | 0.843 | 1.001 | 0.843 |
| | MASE | 0.059 | **0.057** | 0.062 | 0.064 | 0.083 | 0.058 | 509.71 | 1,217.50 | NA | NA |
| NCEI | sMAPE | 0.718 | 0.715 | 0.713 | 0.715 | 0.726 | **0.714** | 0.997 | 0.772 | 1.268 | 1.505 |
| | MASE | 0.593 | **0.591** | 1.011 | 1.039 | 0.712 | **0.591** | 3.722 | 3.614 | 3.573 | 2.979 |
| Runtime (sec) | | 0.26 | 0.80 | 0.32 | 0.97 | 0.16 | 0.68 | **0.04** | 0.05 | 0.58 | 0.50 |

divergence implemented by `GeomLoss` from [16] is utilized, and $\epsilon$ is set to be 0.0025. The Adam optimizer [27] is employed for implementing the optimization given in (3.8). The lookback horizon, forecast horizon, and the number of source domains are set to be $\alpha = 50$, $\beta = 10$, and $K = 3$, respectively (noting that it depends on the characteristics of source domains' datasets). Furthermore, the number of stacks and blocks, and the dimension of feature space are set to be $M = 3$, $L = 4$, and $\gamma = 512$, respectively (noting that it is consistent with N-BEATS [45]). Others are determined through grid search, and the sMAPE and MASE are adopted as evaluation metrics. Additional implementation details and definitions are provided in Appendix D.

**Domain generalization performance.** As shown in Table 1, the proposed stack-wise feature alignment significantly improves the domain shift issue within the deep residual stacking architectures with outstanding performance compared to other benchmarks. In particular, we highlight that the improvement is more significant in ODG where the domain shift from source to target is severely pronounced. That being said, the proposed domain-generalized model can perform and adapt well in a very severe situation without any information on the target environment. Other detailed analysis on the results are discussed in Appendix E.

**Ablation study on divergences.** Table 2 provides the ablation results on the choice of divergence (or distances) for the proposed stack-wise feature alignment, in which the benchmarks consist of the classic (not regularized) Wasserstein-2 distance (WD), the maximum mean discrepancy (MMD), and the Kullback–Leibler divergence (KL) and further sensitivity analysis on the Sinkhorn divergence (SD) with respect to $\epsilon > 0$ is also provided. Due to the heavy running cost for implementing WD cases (see Runtime with 314.30 in Table 2) and the training instability associated with KL cases (see Table 6), we consider the target domain case for 'exchange rate' (within FRED) and the several source domain scenarios for ODG, CDG and IDG (see Appendix D for the details on the source domains' combinations). For the same reasons, the baseline model is fixed to N-BEATS-G. The entire results are provided in Tables 6 and 7.

Table 2: Ablation study on divergences.

| Divergences | | WD | SD ($\epsilon > 0$) | | | MMD | KL |
|---|---|---|---|---|---|---|---|
| | | | 1e-5 | 2.5e-3 | 1e-1 | | |
| ODG | sMAPE | **0.031** | 0.040 | 0.032 | 0.033 | 0.035 | 0.045 |
| | MASE | **0.022** | 0.059 | **0.022** | **0.022** | 0.055 | 0.057 |
| CDG | sMAPE | 0.028 | **0.026** | 0.028 | 0.027 | 0.029 | 0.030 |
| | MASE | **0.039** | 0.058 | 0.040 | **0.039** | 0.041 | **0.039** |
| IDG | sMAPE | **0.024** | **0.024** | **0.024** | 0.025 | 0.025 | 0.026 |
| | MASE | **0.049** | 0.050 | 0.050 | 0.050 | 0.051 | **0.049** |
| Runtime (sec) | | 314.30 | | 0.68 | | 0.81 | **0.53** |

As the Sinkhorn divergence is an accurate approximate of the Wasserstein-2 distance (see Definition 3.5), the similar results for the two cases in Table 2 seem to be reasonable. On the other hand, their computational costs are incomparable. That being said, the Sinkhorn divergence is the computationally feasible and accurate toolkit for the proposed stack-wise alignment with optimal transport based divergence, while some instability issue (see [5, 21]) would come out for extremely small $\epsilon > 0$ (i.e., $\epsilon =$ 1e-5 in Table 2). In comparison with the MMD and KL cases (see Tables 6 and 7 as well for the

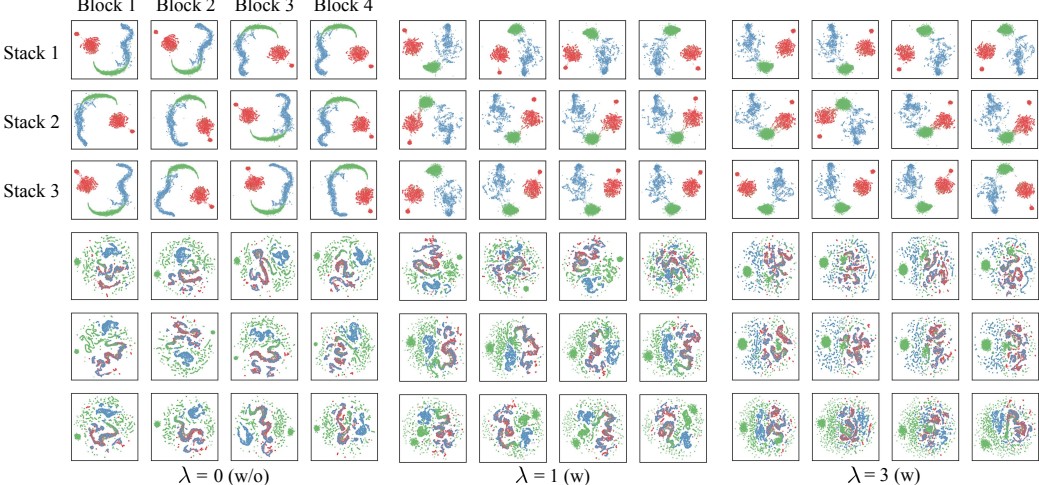

Figure 2: Visualization on invariant feature learning. In the aligned scenario (w), the interconnection between green and red instances, particularly at $\lambda = 3$, becomes visible. Contrastingly, in the non-aligned scenario (w/o), we observe a pronounced dispersion, especially of the blue instances within the initial two stacks at $\lambda = 3$, resulting in heightened inter-domain entropy.

entire results), the Sinkhorn divergence case seems to be marginally better but shows more stable and consistent results in overall domain shift scenarios. From these empirical evidences, we hence conclude that the choice of the Sinkhorn divergence allows the model to bring both the abundant theoretical advantages of optimal transport problems and the practical feasibility.

**Visualization on representation learning.** To visualize representations, i.e., samples of marginal feature measures observed from N-BEATS-G with and without alignment, we use the uniform manifold approximation and projection (UMAP) introduced by [39]. To minimize the effect of unaligned scale information, the softmax function is employed to remove the scale information and instead emphasize the semantic relationship across domains. As illustrated in Figure 2, we observe the proximity between instances and the substantial upsurge in the entropy of domains. For other cases on N-BEATS-I and N-HiTS, please refer to Figure 8 in Appendix F.

**Other results.** On top of the aforementioned results, further experiments are provided in Appendix G, which supports our choices and assumptions on the proposed model. The followings summarize the corresponding results: Comparison of stack-wise and block-wise feature alignment (Appendix G.1); Comparison of several normalization functions (Appendix G.2); Evaluation of the model under marginal (or the absence of) domain shift (Appendix G.3); Evaluation on Tourism [3], M3 [36], and M4 [37] datasets (Appendix G.4). On top of that, we report the train and validation losses in Figure 5 supporting the stable optimization procedure. Furthermore, we provide the visual samples of forecasting results in Figure 6 and make use of the interpretability of the feature-aligned N-BEATS to present Figure 7 (see Appendix F).

## 5   DISCUSSION AND EXTENSIONS

There are some unresolved theoretical parts in the current article such as a convergence analysis for the training loss (given in (3.8)) with the empirical risk minimization and the stack-wise feature alignment, filling the gap between the Sinkhorn divergence and the $\mathcal{H}$-divergence adopted in the error analysis of domain generalization models (given in Proposition 2.1), and the instability issue coming from the small entropic parameter $\epsilon > 0$ in the Sinkhorn divergence (see Table 2).

On the other hand, there are many rooms for an extension of the proposed domain-generalized time series forecasting model such as the 'conditional' feature measure alignment in [65] and 'adversarial representation learning framework' in [30]. Moreover, considering the utilization of 'moments' as distribution measurements in [22] and mitigating distribution mismatches through the 'contrastive loss' in [41] would represent meaningful avenues for future research.

*Acknowledgement.* K. Park gratefully acknowledges support of the Presidential Postdoctoral Fellowship of Nanyang Technological University. M. Kang was supported by the NRF grant [2021R1A2C3010887] and the MSIT/IITP [No. 1711117093; 2021-0-00077; 2021-0-01343, Artificial Intelligence Graduate School Program of SNU].

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

# APPENDIX

## A FURTHER DETAILS ON FEATURE-ALIGNED N-BEATS

The $m$-th residual operators $(\psi^m, \xi_\downarrow^m, \xi_\uparrow^m) \in \Psi \times \Xi_\downarrow \times \Xi_\uparrow$, $m = 1, \dots, M$, are given by

$$
\begin{aligned}
\psi^m(x) &:= \left( \mathrm{FC}^{m,N} \circ \mathrm{FC}^{m,N-1} \cdots \circ \mathrm{FC}^{m,1} \right)(x), & x &\in \mathcal{X} = \mathbb{R}^\alpha, \\
\xi_\downarrow^m(z) &:= \mathbf{V}_{m,\downarrow} \mathbf{W}_{m,\downarrow} z, & z &\in \mathcal{Z} = \mathbb{R}^\gamma, \\
\xi_\uparrow^m(z) &:= \mathbf{V}_{m,\uparrow} \mathbf{W}_{m,\uparrow} z, & z &\in \mathcal{Z} = \mathbb{R}^\gamma.
\end{aligned}
\tag{A.1}
$$

These operators correspond to the $m$-th stack and involve fully connected layers denoted by $\mathrm{FC}^{m,n}$ with RELU activation function [43]. Specifically, $\mathrm{FC}^{m,n}(x)$ is defined as $\mathrm{RELU}(\mathbf{W}_{m,n} x + \mathbf{b}_{m,n})$, where $\mathbf{W}_{m,n}$ and $\mathbf{b}_{m,n}$ are trainable weight and bias parameters, respectively. The matrix $\mathbf{W}_{m,\downarrow} \in \mathbb{R}^{\gamma_\downarrow \times \gamma}$ (resp. $\mathbf{W}_{m,\uparrow} \in \mathbb{R}^{\gamma_\uparrow \times \gamma}$) represents a trainable linear projection layer for forecasting (resp. backcasting) operations. For the parameters $\beta$, $\alpha$, and $\gamma$ denoting to the forecast horizon, lookback horizon, and latent space dimension, respectively, $\mathbf{V}_{m,\downarrow} \in \mathbb{R}^{\beta \times \gamma_\downarrow}$ (resp. $\mathbf{V}_{m,\uparrow} \in \mathbb{R}^{\alpha \times \gamma_\uparrow}$) represents a forecast basis (resp. backcast basis) matrix, given by

$$
\begin{aligned}
\mathbf{V}_{m,\downarrow} &:= (\mathbf{v}_{m,\downarrow}^1, \dots, \mathbf{v}_{m,\downarrow}^{\gamma_\downarrow}) \in \mathbb{R}^{\beta \times \gamma_\downarrow} \quad \text{with} \quad \mathbf{v}_{m,\downarrow}^1, \dots, \mathbf{v}_{m,\downarrow}^{\gamma_\downarrow} \in \mathbb{R}^\beta, \\
(\text{resp.} \quad \mathbf{V}_{m,\uparrow} &:= (\mathbf{v}_{m,\uparrow}^1, \dots, \mathbf{v}_{m,\uparrow}^{\gamma_\uparrow}) \in \mathbb{R}^{\alpha \times \gamma_\uparrow} \quad \text{with} \quad (\mathbf{v}_{m,\uparrow}^1, \dots, \mathbf{v}_{m,\uparrow}^{\gamma_\uparrow}) \in \mathbb{R}^\alpha),
\end{aligned}
\tag{A.2}
$$

and each $\mathbf{v}_{m,\downarrow}^i$ (resp. $\mathbf{v}_{m,\uparrow}^i$), is a forecast basis (resp. backcast basis) vector.

Note that $\mathbf{V}_{m,\downarrow}$ and $\mathbf{V}_{m,\uparrow}$ are set to be non-trainable parameter sets that embrace vital information for time series forecasting purposes, including trends and seasonality. These parameter sets are based on [45]. The basis expansion representations in (A.1) with flexible adjustments in (A.2) allow the model to capture the relevant patterns in the sequential data.

**N-BEATS-G & N-BEATS-I.** The main difference between N-BEATS-G and N-BEATS-I lies on the utilization of $\mathbf{V}_{m,\downarrow}$ and $\mathbf{V}_{m,\uparrow}$. More precisely, N-BEATS-G does not incorporate any specialized time series-specific knowledge but employs the identity matrices for $\mathbf{V}_{m,\downarrow}$ and $\mathbf{V}_{m,\uparrow}$. In contrast, N-BEATS-I captures trend and seasonality information, which derives the interpretability. Specifically, $\mathbf{V}_{m,\downarrow}$ is given by $\mathbf{V}_{m,\downarrow} = (\mathbf{1}, \mathbf{t}, \cdots, \mathbf{t}^{\gamma_\downarrow})$, where $\mathbf{t} = \frac{1}{\beta}(0, 1, 2, \cdots, \beta - 2, \beta - 1)^\top$. This choice is motivated by the characteristic of trends, which are typically represented by monotonic or slowly varying functions. For the seasonality, $\mathbf{V}_{m,\downarrow}$ is defined using a periodic function, (specifically the Fourier series), so that $\mathbf{V}_{m,\downarrow} = (\mathbf{1}, \cos(2\pi \mathbf{t}), \cdots, \cos(2\pi \lfloor \beta/2 - 1 \rfloor \mathbf{t})), \sin(2\pi \mathbf{t}), \cdots, \sin(2\pi \lfloor \beta/2 - 1 \rfloor \mathbf{t}))^\top$. The dimension of $\mathbf{V}_{m,\downarrow}$ is determined by adjusting the interval between $\cos(2\pi \mathbf{t})$ and $\cos(2\pi \lfloor \beta/2 - 1 \rfloor \mathbf{t})$, as well as $\sin(2\pi \mathbf{t})$ and $\sin(2\pi \lfloor \beta/2 - 1 \rfloor \mathbf{t})$. This formulation incorporates the notion of sinusoidal waveforms to capture the periodic nature of seasonality. The formulation of $\mathbf{V}_{m,\uparrow}$ is identical to that of $\mathbf{V}_{m,\downarrow}$, with the only difference being the replacement of $\alpha$ with $\beta$ and $\gamma_\downarrow$ with $\gamma_\uparrow$.

**Lipschitz continuity of residual operators.** Since each $\psi^m$ defined in (A.1) is an $N$-layered fully connected network with the 1-Lipschitz continuous activation, i.e., RELU, we can apply [57, Section 6] to have an explicit representation for the (Rademacher) Lipschitz constant $C_m > 0$ of $\psi^m$ [15, Theorem 3.1.6]. Furthermore, the forecasting and backcasting operators, $\xi_\downarrow^m$ and $\xi_\uparrow^m$, are matrix operators, and we can calculate their Lipschitz constants $C_{m,\downarrow}$ and $C_{m,\uparrow}$ by using the matrix norm induced by the Euclidean norm $\|\cdot\|$, i.e., $C_{m,\downarrow} := \|\mathbf{V}_{m,\downarrow} \mathbf{W}_{m,\downarrow}\| > 0$ and $C_{m,\uparrow} := \|\mathbf{V}_{m,\uparrow} \mathbf{W}_{m,\uparrow}\| > 0$.

**Detailed illustration of stack-wise feature alignment.** In addition to the above-presented N-BEATS, we incorporate the concept of learning invariance for domain generalization, which is referred to as Feature-aligned N-BEATS. We provide the illustration of Feature-aligned N-BEATS in Figure 3 which is a detailed version of Figure 1.

## B PROOFS OF LEMMA 3.4 AND THEOREM 3.6

*Proof of Lemma 3.4.* From the definition of $\sigma \circ g^m$ in Definition 3.1 and the Lipschitz continuity of $\sigma$ and $\psi^m$ with corresponding constants $C_\sigma > 0$ and $C_m > 0$, it follows for every $m = 1, \dots, M$,

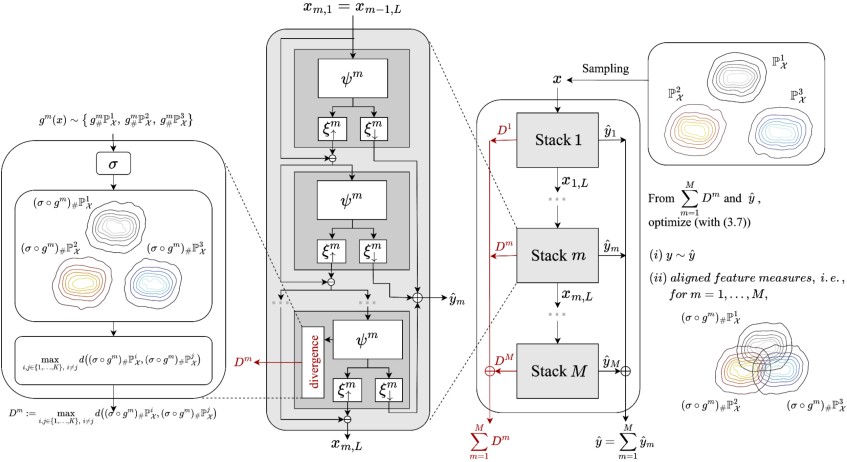

Figure 3: Illustration of Feature-aligned N-BEATS (noting that it is a detailed version of Figure 1).

that for any $x, y \in \mathcal{X}$,

$$
\begin{aligned}
\|\sigma \circ g^m(x) - \sigma \circ g^m(y)\| &\le C_\sigma \|g^m(x) - g^m(y)\| \\
&\le C_\sigma C_m \Big\| \big( (r^m)^{(L-1)} \circ (r^{m-1})^{(L)} \circ \cdots \circ (r^1)^{(L)} \big)(x) \\
&\quad - \big( (r^m)^{(L-1)} \circ (r^{m-1})^{(L)} \circ \cdots \circ (r^1)^{(L)} \big)(y) \Big\|.
\end{aligned}
\tag{B.1}
$$

Based on the residual operation in (3.1), i.e., $r^m(x) = x - (\xi_\uparrow^m \circ \psi^m)(x)$, and considering the Lipschitz continuity of $\sigma$ and $\psi^m$ with respective constants $C_\sigma > 0$ and $C_m > 0$, we can establish the following inequality for every $m = 1, \ldots, M$, and any $x, y \in \mathcal{X}$,

$$
\|r^m(x) - r^m(y)\| \le \|x - y\| + \|(\xi_\uparrow^m \circ \psi^m)(x) - (\xi_\uparrow^m \circ \psi^m)(y)\| \le (1 + C_{m,\uparrow} C_m)\|x - y\|.
$$

Applying this to $L - 1$-times composition of $r^m$, i.e., $(r^m)^{(L-1)}$, we have that for any $x, y \in \mathcal{X}$,

$$
\|(r^m)^{(L-1)}(x) - (r^m)^{(L-1)}(y)\| \le (1 + C_{m,\uparrow} C_m)^{L-1}\|x - y\|.
$$

Using the same arguments for the remaining compositions $(r^{m-1})^{(L)}, (r^{m-2})^{(L)}, \ldots, (r^1)^{(L)}$ in (B.1), we deduce that for any $x, y \in \mathcal{X}$,

$$
\|\sigma \circ g^m(x) - \sigma \circ g^m(y)\| \le C_\sigma C_m (1 + C_m C_{m,\uparrow})^{L-1} \prod_{n=1}^{m-1} (1 + C_n C_{n,\uparrow})^L \|x - y\|.
$$

$\square$

*Proof of Theorem 3.6.* We first note that from the nonnegativity of the entropy term in the regularized Wasserstein distance $\mathcal{W}_{\epsilon, \widetilde{\mathcal{Z}}}$, i.e., for every $\pi \in \Pi(\mu, \nu; \widetilde{\mathcal{Z}})$,

$$
\int_{\widetilde{\mathcal{Z}} \times \widetilde{\mathcal{Z}}} \epsilon \log \left( \frac{d\pi(x, y)}{d\mu(x) d\nu(y)} \right) d\pi(x, y) \ge 0,
$$

it is clear that $\mathcal{W}_{\epsilon, \widetilde{\mathcal{Z}}}(\mu, \nu) \ge 0$ for every $\mu, \nu \in \mathcal{P}(\widetilde{\mathcal{Z}})$. Moreover, from the definition of $\widehat{\mathcal{W}}_{\epsilon, \widetilde{\mathcal{Z}}}$, i.e., $\widehat{\mathcal{W}}_{\epsilon, \widetilde{\mathcal{Z}}}(\mu, \nu) = \mathcal{W}_{\epsilon, \widetilde{\mathcal{Z}}}(\mu, \nu) - \frac{1}{2}(\mathcal{W}_{\epsilon, \widetilde{\mathcal{Z}}}(\nu, \nu) + \mathcal{W}_{\epsilon, \widetilde{\mathcal{Z}}}(\mu, \mu))$, for $\mu, \nu \in \mathcal{P}(\widetilde{\mathcal{Z}})$, it follows that for every $m = 1, \ldots, M$ and any $i, j \in \{1, \ldots, K\}$ such that $i \ne j$,

$$
\widehat{\mathcal{W}}_{\epsilon, \widetilde{\mathcal{Z}}}\big((\sigma \circ g^m)_\# \mathbb{P}_\mathcal{X}^i, (\sigma \circ g^m)_\# \mathbb{P}_\mathcal{X}^j\big) \le \mathcal{W}_{\epsilon, \widetilde{\mathcal{Z}}}\big((\sigma \circ g^m)_\# \mathbb{P}_\mathcal{X}^i, (\sigma \circ g^m)_\# \mathbb{P}_\mathcal{X}^j\big).
\tag{B.2}
$$

Let $\mathcal{C}(\widetilde{\mathcal{Z}})$ be the set of all real-valued continuous functions on $\widetilde{\mathcal{Z}}$ and $\mathcal{C}(\mathcal{X}; \sigma \circ g^m)$ be defined by

$$
\mathcal{C}(\mathcal{X}; \sigma \circ g^m) := \{ f : \mathcal{X} \to \mathbb{R} \mid \exists \widetilde{f} \in \mathcal{C}(\widetilde{\mathcal{Z}}) \text{ s.t. } f = \widetilde{f} \circ \sigma \circ g^m \}.
$$

Then, from the dual representation in [48, Remark 4.18 in Section 4.4] based on the Lagrangian method and the integral property of pushforward measures in [2, Section 5.2], it follows for every $m = 1, \ldots, M$ that given $\mathbb{P}_\mathcal{X}^i, \mathbb{P}_\mathcal{X}^j \in \{\mathbb{P}_\mathcal{X}^k\}_{k=1}^K$ with $i \neq j$,

$$
\begin{aligned}
&\mathcal{W}_{\epsilon, \widetilde{\mathcal{Z}}}\big((\sigma \circ g^m)_\# \mathbb{P}_\mathcal{X}^i, (\sigma \circ g^m)_\# \mathbb{P}_\mathcal{X}^j\big) \\
&= \sup_{\widetilde{f}, \widetilde{h} \in \mathcal{C}(\widetilde{\mathcal{Z}})} \left\{ \int_{\widetilde{\mathcal{Z}}} \widetilde{f}(x) d\big((\sigma \circ g^m)_\# \mathbb{P}_\mathcal{X}^i\big)(x) + \int_{\widetilde{\mathcal{Z}}} \widetilde{h}(y) d\big((\sigma \circ g^m)_\# \mathbb{P}_\mathcal{X}^j\big)(y) \right. \\
&\qquad\left. - \epsilon \int_{\widetilde{\mathcal{Z}} \times \widetilde{\mathcal{Z}}} e^{\frac{1}{\epsilon}(\widetilde{f}(x) + \widetilde{h}(y) - \|x - y\|^2)} d\big((\sigma \circ g^m)_\# \mathbb{P}_\mathcal{X}^i\big) \otimes \big((\sigma \circ g^m)_\# \mathbb{P}_\mathcal{X}^j\big)(x, y) \right\} \\
&= \sup_{f, h \in \mathcal{C}(\mathcal{X}; \sigma \circ g^m)} \left\{ \int_\mathcal{X} f(x) d\mathbb{P}_\mathcal{X}^i(x) + \int_\mathcal{X} h(y) d\mathbb{P}_\mathcal{X}^i(y) \right. \\
&\qquad\left. - \epsilon \int_{\mathcal{X} \times \mathcal{X}} e^{\frac{1}{\epsilon}(f(x) + h(y) - \|(\sigma \circ g^m)(x) - (\sigma \circ g^m)(y)\|^2)} d(\mathbb{P}_\mathcal{X}^i \otimes \mathbb{P}_\mathcal{X}^j)(x, y) \right\}.
\end{aligned}
\tag{B.3}
$$

Consider the following regularized optimal transport problem:

$$
\begin{aligned}
\widetilde{\mathcal{W}}_{\epsilon, \mathcal{X}}(\mathbb{P}^i, \mathbb{P}^j; \sigma \circ g^m) := \inf_{\pi \in \Pi(\mathbb{P}^i, \mathbb{P}^j; \mathcal{X})} &\left\{ \int_{\mathcal{X} \times \mathcal{X}} \Big( \|\sigma \circ g^m(x) - \sigma \circ g^m(y)\|^2 \right. \\
&\left. + \epsilon \log \left( \frac{d\pi(x, y)}{d\mathbb{P}_\mathcal{X}^i(x) d\mathbb{P}_\mathcal{X}^j(y)} \right) \Big) d\pi(x, y) \right\}.
\end{aligned}
$$

Then, from the dual representation, as in (B.3), it follows that

$$
\begin{aligned}
\widetilde{\mathcal{W}}_{\epsilon, \mathcal{X}}(\mathbb{P}^i, \mathbb{P}^j; \sigma \circ g^m) = \sup_{f, h \in \mathcal{C}(\mathcal{X})} &\left\{ \int_\mathcal{X} f(x) d\mathbb{P}_\mathcal{X}^i(x) + \int_\mathcal{X} h(y) d\mathbb{P}_\mathcal{X}^i(y) \right. \\
&\left. - \epsilon \int_{\mathcal{X} \times \mathcal{X}} e^{\frac{1}{\epsilon}(f(x) + h(y) - \|(\sigma \circ g^m)(x) - (\sigma \circ g^m)(y)\|^2)} d(\mathbb{P}_\mathcal{X}^i \otimes \mathbb{P}_\mathcal{X}^j)(x, y) \right\},
\end{aligned}
\tag{B.4}
$$

where $\mathcal{C}(\mathcal{X})$ denotes the set of all continuous real-valued functions on $\mathcal{X}$. From the dual representations in (B.3) and (B.4) and the relation that $\mathcal{C}(\mathcal{X}; \sigma \circ g^m) \subseteq \mathcal{C}(\mathcal{X})$, it follows that

$$
\mathcal{W}_{\epsilon, \widetilde{\mathcal{Z}}}\big((\sigma \circ g^m)_\# \mathbb{P}_\mathcal{X}^i, (\sigma \circ g^m)_\# \mathbb{P}_\mathcal{X}^j\big) \leq \widetilde{\mathcal{W}}_{\epsilon, \mathcal{X}}(\mathbb{P}^i, \mathbb{P}^j; \sigma \circ g^m).
\tag{B.5}
$$

On the other hand, from the first order optimality condition and the continuity of $\sigma \circ g^m$, presented in Lemma 3.4, it follows that the optimal potentials $f^*, h^* \in \mathcal{C}(\mathcal{X})$, which realize the supremum in (B.4), are given by, respectively,

$$
f^*(x) := -\epsilon \log \left( \int_\mathcal{X} e^{\frac{1}{\epsilon}(h^*(y) - \|(\sigma \circ g^m)(x) - (\sigma \circ g^m)(y)\|^2)} d\mathbb{P}_\mathcal{X}^j(y) \right), \quad x \in \mathcal{X},
$$

$$
h^*(y) := -\epsilon \log \left( \int_\mathcal{X} e^{\frac{1}{\epsilon}(f^*(x) - \|(\sigma \circ g^m)(x) - (\sigma \circ g^m)(y)\|^2)} d\mathbb{P}_\mathcal{X}^i(x) \right), \quad y \in \mathcal{X},
$$

which can be represented by $f^* = \widetilde{f}^* \circ \sigma \circ g^m$ and $h^* = \widetilde{h}^* \circ \sigma \circ g^m$, respectively, where $\widetilde{f}^*, \widetilde{h}^* \in \mathcal{C}(\widetilde{\mathcal{Z}})$ are given by, respectively,

$$
\widetilde{f}^*(x) := -\epsilon \log \left( \int_\mathcal{X} e^{\frac{1}{\epsilon}((\widetilde{h}^* \circ \sigma \circ g^m)(y) - \|x - (\sigma \circ g^m)(y)\|^2)} d\mathbb{P}_\mathcal{X}^j(y) \right), \quad x \in \widetilde{\mathcal{Z}},
$$

$$
\widetilde{h}^*(y) := -\epsilon \log \left( \int_\mathcal{X} e^{\frac{1}{\epsilon}((\widetilde{f}^* \circ \sigma \circ g^m)(x) - \|(\sigma \circ g^m)(x) - y\|^2)} d\mathbb{P}_\mathcal{X}^i(x) \right), \quad y \in \widetilde{\mathcal{Z}}.
$$

This ensures that $f^*, h^* \in \mathcal{C}(\mathcal{X}; \sigma \circ g^m) \subseteq \widetilde{\mathcal{C}}(\mathcal{X})$. Hence we establish that (B.5) holds as equality.

From this and the Lipschitz continuity of $\sigma \circ g^m$ with the constant $C_{\sigma \circ g^m} > 0$ in Lemma 3.4, it follows that

$$
\begin{aligned}
\mathcal{W}_{\epsilon, \widetilde{\mathcal{Z}}}\big((\sigma \circ g^m)_{\#}\mathbb{P}_{\mathcal{X}}^i, (\sigma \circ g^m)_{\#}\mathbb{P}_{\mathcal{X}}^j\big) &= \widetilde{\mathcal{W}}_{\epsilon, \mathcal{X}}\big(\mathbb{P}^i, \mathbb{P}^j; \sigma \circ g^m\big) \\
&\leq \inf_{\pi \in \Pi(\mathbb{P}^i, \mathbb{P}^j; \mathcal{X})} \left\{ \int_{\mathcal{X} \times \mathcal{X}} \left( C_{\sigma \circ g^m}^2 \|x - y\|^2 + \epsilon \log \left( \frac{d\pi(x, y)}{d\mathbb{P}_{\mathcal{X}}^i(x) d\mathbb{P}_{\mathcal{X}}^j(y)} \right) \right) d\pi(x, y) \right\} \\
&\leq \max\{C_{\sigma \circ g^m}^2, 1\} \mathcal{W}_{\epsilon, \mathcal{X}}(\mathbb{P}^i, \mathbb{P}^j).
\end{aligned}
$$

Therefore, we have shown that

$$
\sum_{m=1}^{M} \max_{i,j \in \{1,\dots,K\},\ i \neq j} \mathcal{W}_{\epsilon, \widetilde{\mathcal{Z}}}\big((\sigma \circ g^m)_{\#}\mathbb{P}_{\mathcal{X}}^i, (\sigma \circ g^m)_{\#}\mathbb{P}_{\mathcal{X}}^j\big) \leq C \max_{i,j \in \{1,\dots,K\},\ i \neq j} \mathcal{W}_{\epsilon, \mathcal{X}}(\mathbb{P}^i, \mathbb{P}^j),
$$

with $C = \sum_{m=1}^{M} \max\{C_{\sigma \circ g^m}^2, 1\}$. Combining this with (B.2) concludes the proof. $\qquad\square$

## C    SOME REMARKS ON SECTION 3.2

**Remark C.1.** Theorem 3.6 supports that the Sinkhorn-based alignment involving intricate doubly residual stacking architecture is feasible, as far as the pair-wise divergence of source domains' measures can be 'a priori' estimated under some suitable divergence (i.e., the entropic regularized Wasserstein distance in the right-hand side of the inequality therein). Indeed, the `PoT` library introduced by [18] can be used for calculation of the entropic regularized Wasserstein distances. On the other hand, the proposed Sinkhorn-based alignment loss is implemented by `GeomLoss` of [16] that is known to be a significantly efficient and accurate approximate algorithm and will be the main calculation toolkit in the model.

**Remark C.2.** For sequential data generation, [62] introduced a causality constraint within optimal transport distances and used the Sinkhorn divergence as an approximate for the causality constrained optimal transport distances. However, we do not consider the constraint but adopt the Sinkhorn divergence for an approximate of the classic optimal transport distance as in (3.4). This is because unlike the reference, there is no inference for the causality between pushforward feature measures from the source domains.

## D    DETAILED EXPERIMENTAL INFORMATION OF SECTION 4

**Experimental environment.** We conduct all experiments using the specifications below:

- CPU: Intel(R) Xeon(R) Platinum 8163 CPU @ 2.50GHz.
- GPU: NVIDIA TITAN RTX.

All analyses in Section E utilize the same environment. The comprehensive software setup can be found on GitHub[3].

**Dataset configuration.** The training data generation follows the steps detailed below:

1. Retrieve financial data {`commodity`, `income`, `interest rate`, `exchange rate`} from the Federal Reserve Economic Data (FRED), and weather data {`pressure`, `rain`, `temperature`, `wind`} from the National Centers for Environmental Information (NCEI). Subsequently, we designate finance and weather as the superdomain $\mathcal{A}$, with the subordinate data categorized as individual domains.

2. Process each data point into a sliding window of defined dimensions $[\alpha, \beta]$, e.g., [50, 10], with the sliding stride of 1. Each segment is treated as an individual instance.

3. To alleviate any potential concerns arising from data imbalance, we establish a predetermined quantity of 75,000 instances for each domain through random sampling, thereby guaranteeing independence from such considerations.

---

[3] `https://github.com/leejoonhun/fan-beats`

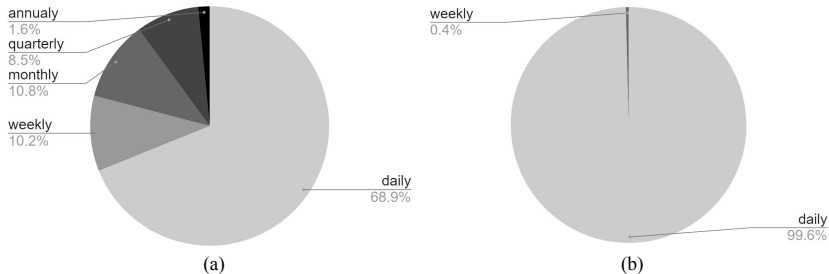

Figure 4: Visualization of frequency distribution. (a) FRED, and (b) NCEI.

4. We randomly split each domain into three sets: 70% for training, 10% for validation, and 20% for testing.

It is worth noting that our dataset consists entirely of real-world data and covers a wide range of frequencies, including daily, weekly, monthly, quarterly, and yearly observations. The graphical representation of the frequency distribution for instances is depicted in Figure 4.

ODG scenarios involve no overlap between the source and target domains with respect to the super-domain, e.g., $\{\mathcal{D}\}_{k=1}^{3} = \{\texttt{pressure}, \texttt{rain}, \texttt{temperature}\}$, and $\mathcal{D}^{T} = \texttt{commodity}$. For fair comparisons, the number of source domains is standardized to three across CDG and IDG scenarios. In CDG, we select one domain from the target domain's superdomain ($p = 2$) and two domains from the other superdomain as sources, e.g., $\{\mathcal{D}\}_{k=1}^{3} = \{\texttt{commodity}, \texttt{income}, \texttt{pressure}\}$, and $\mathcal{D}^{T} = \texttt{rain}$. To evaluate IDG, we designate one target domain and consider the remaining domains within the same superdomain as source domains, e.g., $\{\mathcal{D}\}_{k=1}^{3} = \{\texttt{pressure}, \texttt{rain}, \texttt{temperature}\}$, and $\mathcal{D}^{T} = \texttt{wind}$.

Note that each selected combination of source domains for Table 2 is $\{\texttt{rain}, \texttt{temperature}, \texttt{wind}\}$ for ODG, $\{\texttt{commodity}, \texttt{temperature}, \texttt{wind}\}$ for CDG, and $\{\texttt{commodity}, \texttt{income}, \texttt{interest rate}\}$ for IDG.

**Evaluation metrics.** For our experiments, we employ two evaluation metrics. Given $H = N \times \beta$ with where $N$ represents the number of instances, the metrics are defined as:

$$\text{sMAPE} = \frac{2}{H} \times \sum_{i=1}^{H} \frac{|y_i - \hat{y}_i|}{|y_i| + |\hat{y}_i|}, \quad \text{and} \quad \text{MASE} = \frac{1/H \times \sum_{i=1}^{H} |y_i - \hat{y}_i|}{1/(H-1) \times \sum_{i=1}^{H-1} |y_{i+1} - y_i|}.$$

**Hyperparameters.** Considering the scope of our experimental configuration that bring a total of 184 experimental cases for each model, we adopt a suitable range of hyperparameters, detailed in Table 3, to achieve the performance results presented in Table 1.

**Training and validation loss plots.** Feature-aligned N-BEATS is a complicated architecture based on the doubly residual stacking principle with an additional feature-aligning procedure. To investigate the stability of training, we analyze the training and validation loss plots. Figure 5 indicates that the gradients are well-behaved during training. This stable optimization is regarded to the Lemma 3.4 presented in Section 3.1.

Table 3: Hyperparameters.

| Hyperparameter | Considered Values |
|---|---|
| Lookback horizon. | $\alpha = 50$ |
| Forecast horizon. | $\beta = 10$ |
| Number of stacks. | $M = 3$ |
| Number of blocks in each stack. | $L = 4$ |
| Activation function. | ReLu |
| Feature dimension. | $\gamma = 512$ |
| Loss function. | $\mathcal{L} = $ sMAPE |
| Regularizing temperature. | $\lambda \in \{0.1, 0.3, 1, 3\}$ |
| Learning rate scheduling. | `CyclicLR(base_lr=2e-7, max_lr=2e-5,`
`mode="triangular2",`
`step_size_up=10)` |
| Batch size. | $B = 2^{12}$ |
| Number of iterations. | 1,000 |
| Type of stacks used for N-BEATS-I. | [Trend, Seasonality, Seasonality] |
| Number of polynomials and harmonics used for N-BEATS-I. | 2 |
| Pooling method used for N-HiTS. | `MaxPool1d` |
| Interpolation method used for N-HiTS. | `interpolate(mode="linear")` |
| Kernel size used for N-HiTS. | 2 |

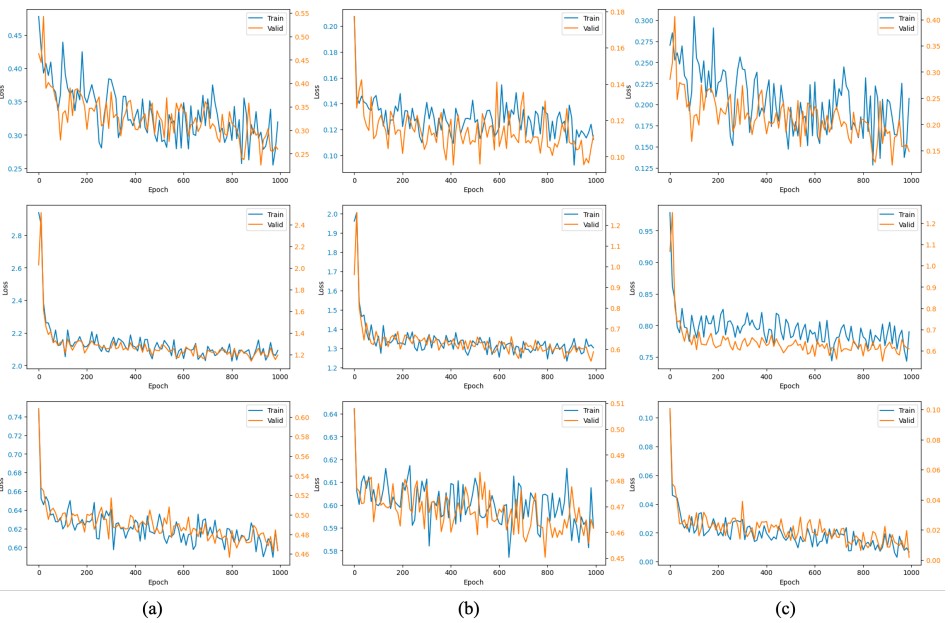

(a)  (b)  (c)

Figure 5: Training and validation loss plots. (a) Total loss, (b) forecasting loss, and (c) alignment loss. From top to bottom, each row illustrates the losses of N-BEATS-G, N-BEATS-I, and N-HiTS, respectively. Losses are reported every 10 iterations.

# E    DETAILED EXPERIMENTAL RESULTS OF SECTION 4

Tables 4 and 5 contain the extended experimental results summarized in Table 1. The suitability of various measures of dispersion within our proposed framework explored in Table 2 is presented in Tables 6 and 7 with more details. Specifically, we consider the commonly used metrics for the representation learning framework: for $\mu, \nu \in \mathcal{P}(\widetilde{\mathcal{Z}})$,

- Kullback-Leibler divergence (KL):

$$\mathrm{KL}(\mu\|\nu) = \int_{\widetilde{\mathcal{Z}}} \log\left(\frac{\mu(dz)}{\nu(dz)}\right)\mu(dz),$$

- Maximum mean discrepancy (MMD):

$$\mathrm{MMD}_{\mathcal{F}}(\mu, \nu) = \sup_{f\in\mathcal{F}}\left(\int_{\widetilde{\mathcal{Z}}} f(x)\mu(dx) - \int_{\widetilde{\mathcal{Z}}} f(y)\nu(dy)\right),$$

where $\mathcal{F}$ represents a class of functions $f : \widetilde{\mathcal{Z}} \to \mathbb{R}$. Notably, $\mathcal{F}$ can be delineated as the unit ball in a reproducing kernel Hilbert space. For detailed description and insights into other possible function classes, refer to [23, Sections 2.2, 7.1, and 7.2].

Table 4: Domain generalization performance of Feature-aligned N-BEATS (noting that it represents entire stats corresponding to the averaged ones given in Table 1). The first two columns denote the target domain for each experiment. This is followed in subsequent tables as well.

| | Methods | N-HiTS | | + FA (Ours) | | N-BEATS-I | | + FA (Ours) | | N-BEATS-G | | + FA (Ours) | |
|---|---|---|---|---|---|---|---|---|---|---|---|---|---|
| | Metrics | sMAPE | MASE | sMAPE | MASE | sMAPE | MASE | sMAPE | MASE | sMAPE | MASE | sMAPE | MASE |
| | | | | | | | ODG | | | | | | |
| FRED | Commodity | 0.136 | 0.049 | **0.103** | **0.046** | 0.279 | 0.072 | 0.258 | 0.069 | 0.195 | 0.052 | 0.136 | 0.049 |
| | Income | 0.299 | 0.057 | **0.298** | **0.055** | 0.369 | 0.082 | 0.335 | 0.075 | 0.305 | 0.056 | 0.304 | **0.055** |
| | Interest rate | 0.120 | 0.074 | **0.100** | 0.070 | 0.200 | **0.044** | 0.189 | 0.046 | 0.148 | 0.073 | 0.120 | 0.073 |
| | Exchange rate | 0.035 | 0.059 | **0.034** | **0.058** | 0.078 | 0.078 | 0.075 | 0.069 | 0.040 | 0.062 | 0.039 | 0.060 |
| | Average | 0.148 | 0.060 | **0.134** | **0.057** | 0.232 | 0.069 | 0.214 | 0.065 | 0.172 | 0.061 | 0.150 | 0.059 |
| NCEI | Pressure | 0.368 | 0.255 | **0.350** | **0.250** | 0.759 | 0.396 | 0.409 | 0.305 | 0.368 | 0.255 | 0.367 | 0.255 |
| | Rain | 1.821 | 1.099 | 1.804 | **0.910** | 1.798 | 1.600 | **1.793** | 1.430 | 1.818 | 1.100 | 1.806 | 0.918 |
| | Temperature | 0.247 | 0.245 | **0.245** | **0.243** | 0.247 | 0.353 | 0.246 | 0.255 | 0.247 | 0.245 | 0.246 | 0.244 |
| | Wind | 0.455 | 0.645 | 0.454 | **0.644** | 0.451 | 0.665 | **0.449** | 0.662 | 0.455 | 0.645 | 0.454 | 0.645 |
| | Average | 0.723 | 0.561 | **0.713** | **0.512** | 0.814 | 0.754 | 0.724 | 0.663 | 0.722 | 0.561 | 0.718 | 0.516 |
| | | | | | | | CDG | | | | | | |
| FRED | Commodity | 0.082 | 0.047 | **0.081** | **0.044** | 0.195 | 0.060 | 0.197 | 0.059 | 0.119 | 0.046 | 0.107 | 0.046 |
| | Income | 0.296 | 0.054 | **0.295** | **0.053** | 0.327 | 0.072 | 0.323 | 0.070 | 0.301 | 0.055 | **0.295** | **0.053** |
| | Interest rate | 0.086 | 0.074 | **0.085** | 0.074 | 0.146 | **0.051** | 0.146 | 0.054 | 0.102 | 0.077 | 0.100 | 0.077 |
| | Exchange rate | 0.032 | 0.056 | **0.029** | **0.055** | 0.054 | 0.074 | 0.055 | 0.063 | 0.032 | 0.056 | 0.031 | **0.055** |
| | Average | 0.124 | 0.058 | **0.123** | **0.057** | 0.181 | 0.064 | 0.179 | 0.062 | 0.139 | 0.059 | 0.133 | 0.058 |
| NCEI | Pressure | 0.375 | 0.264 | **0.372** | **0.260** | 0.408 | 0.307 | 0.405 | 0.299 | 0.540 | 0.374 | **0.372** | 0.263 |
| | Rain | 1.807 | 1.169 | 1.803 | **0.783** | 1.800 | 2.091 | **1.787** | 1.831 | 1.807 | 1.169 | 1.804 | 1.178 |
| | Temperature | 0.334 | 0.243 | 0.245 | **0.242** | 0.275 | 0.245 | **0.240** | 0.244 | 0.253 | 0.245 | 0.245 | 0.243 |
| | Wind | 0.453 | **0.643** | 0.452 | **0.643** | 0.441 | 0.646 | **0.439** | 0.646 | 0.453 | **0.643** | 0.452 | **0.643** |
| | Average | 0.742 | 0.581 | **0.718** | **0.482** | 0.731 | 0.822 | **0.718** | 0.755 | 0.763 | 0.608 | 0.718 | 0.582 |
| | | | | | | | IDG | | | | | | |
| FRED | Commodity | 0.083 | 0.045 | **0.068** | **0.043** | 0.125 | 0.053 | 0.126 | 0.053 | 0.165 | 0.047 | 0.080 | 0.044 |
| | Income | 0.299 | 0.058 | **0.297** | **0.054** | 0.305 | 0.055 | 0.302 | 0.055 | 0.301 | 0.056 | 0.298 | 0.055 |
| | Interest rate | 0.072 | **0.081** | 0.071 | **0.081** | 0.088 | 0.084 | 0.086 | 0.091 | 0.080 | 0.083 | 0.074 | **0.081** |
| | Exchange rate | **0.024** | 0.051 | **0.024** | **0.050** | 0.028 | 0.056 | 0.029 | 0.056 | 0.025 | 0.051 | **0.024** | **0.050** |
| | Average | 0.119 | 0.059 | **0.115** | **0.057** | 0.137 | 0.062 | 0.136 | 0.064 | 0.143 | 0.081 | 0.119 | 0.058 |
| NCEI | Pressure | 0.394 | 0.276 | **0.384** | 0.272 | 0.392 | 0.266 | 0.389 | **0.263** | **0.384** | 0.276 | **0.384** | 0.276 |
| | Rain | **1.776** | 1.211 | **1.776** | **1.208** | 1.792 | 2.922 | 1.805 | 3.046 | 1.818 | 1.690 | **1.776** | **1.208** |
| | Temperature | 0.247 | 0.242 | 0.247 | 0.242 | 0.234 | 0.231 | **0.231** | **0.227** | 0.247 | 0.242 | 0.245 | 0.241 |
| | Wind | 0.455 | 0.641 | 0.452 | 0.640 | **0.433** | **0.622** | 0.434 | 0.623 | 0.454 | 0.641 | 0.452 | 0.640 |
| | Average | 0.718 | 0.593 | 0.715 | **0.591** | **0.713** | 1.011 | 0.715 | 1.039 | 0.726 | 0.712 | 0.714 | **0.591** |

Table 5: Domain generalization performance of competing models (noting that it represents entire stats corresponding to the averaged ones given in Table 1). Note that 'NA' indicates an anomalous error exceeding 10,000, not a divergence of training, as in Table 1.

| | Methods | NLinear | | DLinear | | Autoformer | | Informer | |
|---|---|---|---|---|---|---|---|---|---|
| | Metrics | sMAPE | MASE | sMAPE | MASE | sMAPE | MASE | sMAPE | MASE |
| | ODG | | | | | | | | |
| FRED | Commodity | 0.666 | **17.941** | **0.657** | 18.084 | 0.830 | 22.325 | 1.248 | 40.580 |
| | Income | **0.003** | **162.32** | 0.185 | 8,752.66 | 0.762 | NA | 1.275 | NA |
| | Interest rate | **0.022** | **9.138** | 0.190 | 83.204 | 0.363 | 144.97 | 1.209 | 2,184.78 |
| | Exchange rate | **0.013** | **3.189** | 0.196 | 3.979 | 0.326 | 4.531 | 1.124 | 27.990 |
| | Average | **0.176** | **48.147** | 0.307 | 2,214.48 | 0.570 | NA | 1.214 | NA |
| NCEI | Pressure | **0.954** | **3.837** | 1.300 | 4.237 | 1.168 | 4.749 | 1.414 | 8.284 |
| | Rain | **1.038** | 1.089 | 1.231 | 1.169 | 1.175 | **0.897** | 1.783 | 1.162 |
| | Temperature | **1.136** | **4.399** | 1.340 | 4.572 | 1.352 | 5.761 | 1.616 | 10.885 |
| | Wind | **1.320** | 1.623 | 1.337 | **1.497** | 1.476 | 1.836 | 1.706 | 2.803 |
| | Average | **1.112** | **2.737** | 1.302 | 2.869 | 1.293 | 3.311 | 1.630 | 5.784 |
| | CDG | | | | | | | | |
| FRED | Commodity | **0.664** | **18.166** | 0.855 | 21.661 | 0.829 | 21.360 | 1.353 | 40.584 |
| | Income | **0.004** | **212.60** | 0.203 | 9,979.14 | 0.995 | NA | 1.204 | NA |
| | Interest rate | **0.023** | **9.364** | 0.587 | 210.91 | 0.957 | 1,695.19 | 1.040 | 2,058.07 |
| | Exchange rate | **0.014** | **3.586** | 0.499 | 5.352 | 0.792 | 14.055 | 0.974 | 28.157 |
| | Average | **0.176** | **60.929** | 0.536 | 2,554.27 | 0.893 | NA | 1.143 | NA |
| NCEI | Pressure | **0.923** | **3.810** | 1.055 | 4.100 | 1.127 | 4.769 | 1.222 | 5.851 |
| | Rain | 1.037 | 1.137 | **1.033** | 1.125 | 1.201 | **0.899** | 1.523 | 1.149 |
| | Temperature | 1.114 | **4.340** | **1.106** | 4.431 | 1.304 | 5.609 | 1.439 | 7.358 |
| | Wind | 1.310 | 1.649 | **1.151** | **1.491** | 1.458 | 1.655 | 1.565 | 2.228 |
| | Average | 1.096 | **2.734** | **1.086** | 2.787 | 1.273 | 3.233 | 1.437 | 4.147 |
| | IDG | | | | | | | | |
| FRED | Commodity | **0.671** | 30.210 | 1.139 | 38.929 | 0.835 | **21.462** | 1.652 | 34.784 |
| | Income | **0.026** | **1,951.16** | 0.043 | 4,232.97 | 1.500 | NA | 0.704 | NA |
| | Interest rate | **0.079** | **52.007** | 1.111 | 586.24 | 0.943 | 475.12 | 0.593 | 333.18 |
| | Exchange rate | **0.013** | 5.443 | 1.080 | 11.873 | 0.726 | 5.815 | 0.424 | **5.255** |
| | Average | **0.197** | **509.71** | 0.843 | 1,217.50 | 1.001 | NA | 0.843 | NA |
| NCEI | Pressure | 0.868 | 5.271 | **0.698** | 5.318 | 1.280 | 6.992 | 1.906 | **4.614** |
| | Rain | 0.900 | 1.368 | **0.709** | 1.301 | 1.009 | **0.918** | 1.463 | 1.081 |
| | Temperature | 1.014 | 6.055 | **0.768** | 5.753 | 1.310 | 4.788 | 1.080 | **4.235** |
| | Wind | 1.206 | 2.195 | **0.913** | 2.084 | 1.472 | **1.592** | 1.570 | 1.986 |
| | Average | 0.997 | 3.722 | **0.772** | 3.614 | 1.268 | 3.573 | 1.505 | **2.979** |

Table 6: Ablation study on other divergences (noting that it represents entire stats containing the ones given in Table 2)

| Models | | N-HiTS | | | | | | N-BEATS-I | | | | | | N-BEATS-G | | | | | |
|---|---|---|---|---|---|---|---|---|---|---|---|---|---|---|---|---|---|---|---|
| Divergences | | WD | | MMD | | KL | | WD | | MMD | | KL | | WD | | MMD | | KL | |
| Metrics | | sMAPE | MASE | sMAPE | MASE | sMAPE | MASE | sMAPE | MASE | sMAPE | MASE | sMAPE | MASE | sMAPE | MASE | sMAPE | MASE | sMAPE | MASE |
| ODG | | | | | | | | | | | | | | | | | | | |
| FRED | Commodity | 0.103 | **0.046** | 0.110 | **0.046** | **0.100** | 0.047 | 0.257 | 0.069 | 0.246 | 0.066 | 0.222 | 0.063 | 0.136 | 0.049 | 0.137 | 0.050 | 0.133 | 0.050 |
| FRED | Income | 0.297 | 0.055 | 0.302 | 0.056 | **0.293** | **0.050** | 0.334 | 0.075 | 0.338 | 0.082 | 0.320 | 0.067 | 0.305 | 0.055 | 0.305 | 0.055 | 0.300 | 0.052 |
| FRED | Interest rate | **0.100** | 0.070 | 0.107 | 0.083 | **0.100** | 0.070 | 0.189 | 0.046 | 0.165 | 0.021 | 0.189 | 0.046 | 0.119 | 0.073 | 0.123 | 0.077 | 0.120 | 0.073 |
| FRED | Exchange rate | **0.034** | 0.058 | **0.034** | 0.058 | 0.040 | **0.053** | 0.075 | 0.069 | 0.074 | 0.073 | 0.070 | 0.070 | 0.039 | 0.060 | 0.041 | 0.059 | 0.044 | 0.057 |
| FRED | Average | 0.134 | 0.057 | 0.138 | 0.061 | **0.133** | **0.055** | 0.214 | 0.065 | 0.206 | 0.061 | 0.200 | 0.062 | 0.150 | 0.059 | 0.152 | 0.060 | 0.149 | 0.058 |
| NCEI | Pressure | **0.349** | **0.250** | 0.351 | **0.250** | NaN | NaN | 0.411 | 0.305 | 0.412 | 0.306 | NaN | NaN | 0.367 | 0.255 | 0.367 | 0.254 | 0.365 | 0.263 |
| NCEI | Rain | 1.807 | **0.910** | 1.820 | 0.919 | NaN | NaN | **1.789** | 1.430 | 1.800 | 1.728 | NaN | NaN | 1.798 | 0.918 | 1.818 | 1.026 | 1.831 | 0.951 |
| NCEI | Temperature | **0.245** | **0.243** | 0.246 | 0.244 | NaN | NaN | 0.246 | 0.255 | 0.248 | 0.255 | NaN | NaN | 0.246 | 0.244 | 0.248 | 0.245 | 0.248 | 0.247 |
| NCEI | Wind | 0.454 | **0.644** | 0.455 | 0.645 | NaN | NaN | **0.450** | 0.662 | **0.450** | 0.662 | NaN | NaN | 0.454 | 0.645 | 0.457 | 0.645 | 0.456 | 0.646 |
| NCEI | Average | **0.714** | **0.512** | 0.718 | 0.515 | NaN | NaN | 0.724 | 0.663 | 0.728 | 0.738 | NaN | NaN | 0.716 | 0.515 | 0.723 | 0.543 | 0.725 | 0.527 |
| CDG | | | | | | | | | | | | | | | | | | | |
| FRED | Commodity | **0.081** | **0.044** | 0.086 | 0.045 | NaN | NaN | 0.197 | 0.059 | 0.187 | 0.057 | NaN | NaN | 0.107 | 0.046 | 0.104 | 0.046 | NaN | NaN |
| FRED | Income | **0.295** | 0.053 | 0.298 | 0.054 | NaN | NaN | 0.322 | 0.070 | 0.323 | 0.071 | 0.315 | 0.064 | 0.296 | 0.053 | 0.301 | 0.055 | 0.296 | **0.051** |
| FRED | Interest rate | **0.085** | 0.074 | 0.088 | 0.076 | NaN | NaN | 0.146 | 0.054 | 0.141 | 0.051 | 0.115 | **0.035** | 0.100 | 0.077 | 0.100 | 0.078 | 0.088 | 0.076 |
| FRED | Exchange rate | **0.029** | 0.055 | **0.029** | 0.056 | 0.030 | **0.055** | 0.055 | 0.063 | 0.054 | 0.066 | 0.044 | 0.071 | 0.031 | 0.055 | 0.031 | 0.057 | 0.031 | 0.057 |
| FRED | Average | **0.122** | **0.056** | 0.125 | 0.058 | NaN | NaN | 0.180 | 0.061 | 0.176 | 0.061 | NaN | NaN | 0.133 | 0.058 | 0.134 | 0.059 | NaN | NaN |
| NCEI | Pressure | 0.372 | **0.260** | 0.377 | 0.262 | NaN | NaN | 0.403 | 0.299 | 0.405 | 0.316 | NaN | NaN | 0.371 | 0.263 | **0.370** | 0.262 | NaN | NaN |
| NCEI | Rain | 1.796 | **0.783** | 1.815 | 0.940 | NaN | NaN | **1.780** | 1.831 | 1.791 | 1.934 | NaN | NaN | 1.804 | 1.178 | 1.808 | 1.108 | NaN | NaN |
| NCEI | Temperature | 0.244 | **0.242** | 0.245 | 0.243 | NaN | NaN | **0.240** | 0.244 | 0.241 | 0.245 | NaN | NaN | 0.244 | 0.243 | 0.246 | 0.244 | NaN | NaN |
| NCEI | Wind | 0.452 | **0.643** | 0.453 | **0.643** | NaN | NaN | **0.437** | 0.646 | 0.440 | 0.646 | NaN | NaN | 0.453 | **0.643** | 0.453 | **0.643** | NaN | NaN |
| NCEI | Average | 0.716 | **0.482** | 0.723 | 0.522 | NaN | NaN | **0.715** | 0.755 | 0.719 | 0.785 | NaN | NaN | 0.718 | 0.582 | 0.719 | 0.564 | NaN | NaN |
| IDG | | | | | | | | | | | | | | | | | | | |
| FRED | Commodity | **0.068** | **0.043** | **0.068** | 0.044 | NaN | NaN | 0.126 | 0.053 | 0.122 | 0.050 | NaN | NaN | **0.080** | 0.044 | 0.082 | 0.045 | NaN | NaN |
| FRED | Income | **0.297** | **0.054** | 0.299 | 0.057 | NaN | NaN | 0.302 | 0.055 | 0.308 | 0.058 | NaN | NaN | **0.297** | 0.055 | 0.301 | 0.056 | NaN | NaN |
| FRED | Interest rate | **0.071** | 0.081 | 0.072 | **0.080** | NaN | NaN | 0.086 | 0.091 | 0.098 | 0.089 | NaN | NaN | 0.074 | 0.081 | 0.074 | 0.082 | NaN | NaN |
| FRED | Exchange rate | **0.024** | 0.050 | 0.025 | 0.051 | NaN | NaN | 0.029 | 0.056 | 0.035 | 0.055 | NaN | NaN | **0.024** | 0.050 | 0.025 | 0.051 | 0.026 | **0.049** |
| FRED | Average | **0.115** | **0.057** | 0.116 | 0.058 | NaN | NaN | 0.136 | 0.064 | 0.141 | 0.063 | NaN | NaN | 0.119 | 0.058 | 0.121 | 0.059 | NaN | NaN |
| NCEI | Pressure | 0.384 | 0.272 | 0.403 | 0.286 | 0.403 | 0.286 | 0.390 | 0.263 | 0.384 | **0.259** | 0.380 | 0.261 | 0.382 | 0.276 | **0.376** | 0.274 | NaN | NaN |
| NCEI | Rain | 1.782 | **1.208** | 1.849 | 1.421 | NaN | NaN | 1.800 | 3.045 | 1.792 | 2.881 | NaN | NaN | **1.767** | **1.208** | 1.817 | 1.687 | NaN | NaN |
| NCEI | Temperature | 0.248 | 0.242 | 0.247 | 0.242 | NaN | NaN | **0.230** | **0.227** | 0.234 | 0.228 | NaN | NaN | 0.245 | 0.241 | 0.245 | 0.242 | NaN | NaN |
| NCEI | Wind | 0.453 | 0.640 | 0.454 | 0.640 | NaN | NaN | **0.433** | **0.623** | 0.435 | 0.624 | NaN | NaN | 0.451 | 0.640 | 0.452 | 0.641 | NaN | NaN |
| NCEI | Average | 0.717 | **0.591** | 0.738 | 0.647 | NaN | NaN | 0.713 | 1.039 | **0.711** | 0.998 | NaN | NaN | **0.711** | **0.591** | 0.723 | 0.711 | NaN | NaN |

Table 7: Ablation study on the Sinkhorn divergence with several values on $\epsilon$ (noting that it represents entire stats containing the ones given in Table 2).

| Models | | N-HiTS | | | | N-BEATS-I | | | | N-BEATS-G | | | |
|---|---|---|---|---|---|---|---|---|---|---|---|---|---|
| $\epsilon$ Values | | 1e-5 | | 1e-1 | | 1e-5 | | 1e-1 | | 1e-5 | | 1e-1 | |
| Metrics | | sMAPE | MASE | sMAPE | MASE | sMAPE | MASE | sMAPE | MASE | sMAPE | MASE | sMAPE | MASE |
| ODG | | | | | | | | | | | | | |
| FRED | Commodity | **0.103** | **0.046** | 0.109 | 0.047 | 0.257 | 0.069 | 0.121 | 0.097 | 0.110 | 0.047 | 0.112 | 0.074 |
| FRED | Income | **0.297** | **0.055** | **0.297** | 0.129 | 0.334 | 0.075 | 0.329 | 0.267 | 0.302 | 0.057 | 0.304 | 0.204 |
| FRED | Interest rate | **0.100** | 0.070 | 0.105 | **0.046** | 0.189 | **0.046** | 0.116 | 0.094 | 0.106 | 0.074 | 0.107 | 0.072 |
| FRED | Exchange rate | **0.034** | 0.058 | 0.036 | **0.015** | 0.075 | 0.069 | 0.040 | 0.031 | 0.036 | 0.060 | 0.037 | 0.024 |
| FRED | Average | **0.134** | **0.057** | 0.137 | 0.059 | 0.214 | 0.065 | 0.152 | 0.122 | 0.139 | 0.060 | 0.140 | 0.094 |
| NCEI | Pressure | **0.348** | 0.250 | 0.356 | **0.156** | 0.408 | 0.305 | 0.393 | 0.322 | 0.364 | 0.254 | 0.364 | 0.246 |
| NCEI | Rain | 1.795 | 0.910 | 1.790 | **0.776** | **1.789** | 1.430 | 1.980 | 1.603 | 1.808 | 0.944 | 1.832 | 1.223 |
| NCEI | Temperature | **0.244** | 0.243 | 0.246 | **0.106** | 0.245 | 0.255 | 0.272 | 0.219 | 0.247 | 0.244 | 0.252 | 0.167 |
| NCEI | Wind | 0.452 | 0.644 | 0.450 | **0.195** | **0.448** | 0.662 | 0.498 | 0.404 | 0.457 | 0.645 | 0.461 | 0.308 |
| NCEI | Average | **1.072** | 0.580 | 1.073 | **0.466** | 1.099 | 0.868 | 1.187 | 0.963 | 1.086 | 0.599 | 1.098 | 0.735 |
| CDG | | | | | | | | | | | | | |
| FRED | Commodity | **0.081** | **0.044** | 0.087 | 0.057 | 0.197 | 0.059 | 0.093 | 0.085 | 0.087 | 0.045 | 0.088 | 0.066 |
| FRED | Income | **0.295** | **0.053** | 0.298 | 0.193 | 0.323 | 0.070 | 0.318 | 0.290 | 0.298 | 0.056 | 0.302 | 0.225 |
| FRED | Interest rate | **0.085** | 0.074 | 0.090 | 0.058 | 0.146 | **0.054** | 0.096 | 0.086 | 0.090 | 0.078 | 0.091 | 0.067 |
| FRED | Exchange rate | **0.029** | 0.055 | 0.030 | **0.020** | 0.055 | 0.063 | 0.032 | 0.030 | 0.030 | 0.057 | 0.030 | 0.023 |
| FRED | Average | **0.123** | **0.057** | 0.126 | 0.082 | 0.180 | 0.062 | 0.135 | 0.123 | 0.126 | 0.059 | 0.128 | 0.095 |
| NCEI | Pressure | 0.372 | 0.260 | 0.369 | **0.240** | 0.405 | 0.299 | 0.393 | 0.361 | **0.367** | 0.261 | 0.374 | 0.280 |
| NCEI | Rain | 1.803 | **0.783** | 1.790 | 1.163 | **1.787** | 1.831 | 1.910 | 1.747 | 1.801 | 0.965 | 1.816 | 1.355 |
| NCEI | Temperature | 0.245 | 0.242 | 0.245 | **0.159** | **0.240** | 0.244 | 0.261 | 0.239 | 0.246 | 0.244 | 0.248 | 0.185 |
| NCEI | Wind | 0.452 | 0.643 | 0.451 | **0.293** | **0.439** | 0.646 | 0.481 | 0.440 | 0.454 | 0.643 | 0.457 | 0.341 |
| NCEI | Average | 1.088 | **0.522** | **1.080** | 0.702 | 1.096 | 1.065 | 1.152 | 1.054 | 1.084 | 0.613 | 1.095 | 0.818 |
| IDG | | | | | | | | | | | | | |
| FRED | Commodity | **0.068** | **0.043** | 0.070 | 0.056 | 0.126 | 0.053 | 0.071 | 0.092 | 0.070 | 0.044 | 0.070 | 0.055 |
| FRED | Income | **0.297** | **0.054** | 0.304 | 0.240 | 0.302 | 0.055 | 0.308 | 0.397 | 0.299 | 0.058 | 0.303 | 0.238 |
| FRED | Interest rate | **0.071** | 0.081 | 0.072 | 0.058 | 0.086 | 0.091 | 0.073 | 0.095 | 0.073 | 0.085 | 0.072 | **0.057** |
| FRED | Exchange rate | **0.024** | 0.050 | 0.025 | **0.020** | 0.029 | 0.056 | 0.025 | 0.033 | 0.025 | 0.050 | 0.025 | **0.020** |
| FRED | Average | **0.115** | **0.057** | 0.118 | 0.094 | 0.136 | 0.064 | 0.119 | 0.154 | 0.117 | 0.059 | 0.118 | 0.093 |
| NCEI | Pressure | **0.384** | 0.273 | 0.389 | 0.310 | 0.389 | **0.263** | 0.395 | 0.512 | 0.386 | 0.277 | 0.388 | 0.307 |
| NCEI | Rain | **1.776** | **1.212** | 1.785 | 1.422 | 1.805 | 3.046 | 1.811 | 2.348 | 1.781 | 1.326 | 1.781 | 1.409 |
| NCEI | Temperature | 0.247 | 0.243 | 0.247 | 0.196 | **0.231** | 0.227 | 0.250 | 0.323 | 0.246 | 0.241 | 0.246 | **0.194** |
| NCEI | Wind | 0.452 | 0.642 | 0.452 | 0.360 | **0.434** | **0.623** | 0.459 | 0.595 | 0.452 | 0.640 | 0.451 | **0.357** |
| NCEI | Average | **1.080** | **0.743** | 1.087 | 0.866 | 1.097 | 1.655 | 1.103 | 1.430 | 1.084 | 0.802 | 1.085 | 0.858 |

# F  VISUALIZATION ON FORECASTING, INTERPRETABILITY AND REPRESENTATION

**Visual comparison of forecasts.** We visually compare our models to the N-BEATS-based models, i.e., N-BEATS-G, N-BEATS-I, and N-HiTS. As illustrated in Figure 6, incorporating feature alignment remarkably enhances generalizability, allowing the models to produce finer forecast details. Notably, while baseline models suffer significant performance degradation in the ODG and CDG scenarios, Feature-aligned N-BEATS evidences the benefits of the feature alignment.

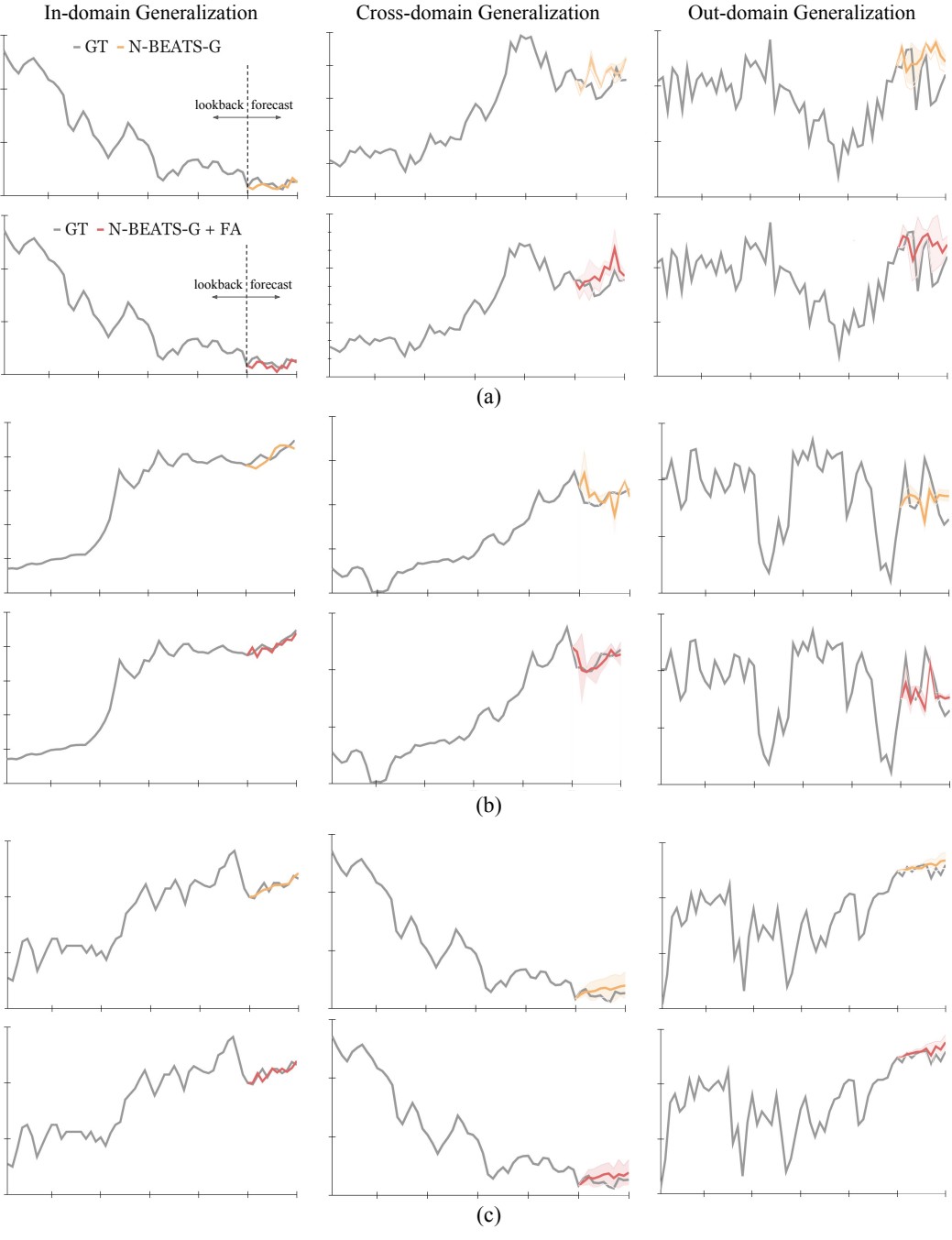

Figure 6:   Visual comparison of forecasts.  (a) N-BEATS-G, (b) N-BEATS-I, and (c) N-HiTS. Results are averaged across source domain combinations, with standard deviations.

**Visual analysis of interpretability.** Figure 7 exhibits the interpretability of the proposed method by presenting the final output of the model and intermediate stack forecasts. N-BEATS-I and N-HiTS presented in Appendix A have interpretability. More specifically, N-BEATS-I explicitly captures trend and seasonality information using polynomial and harmonic basis functions, respectively. N-HiTS employs Fourier decomposition and utilizes its stacks for hierarchical forecasting based on frequencies. Preserving these core architectures during the alignment procedure, Feature-aligned N-BEATS still retains interpretability.

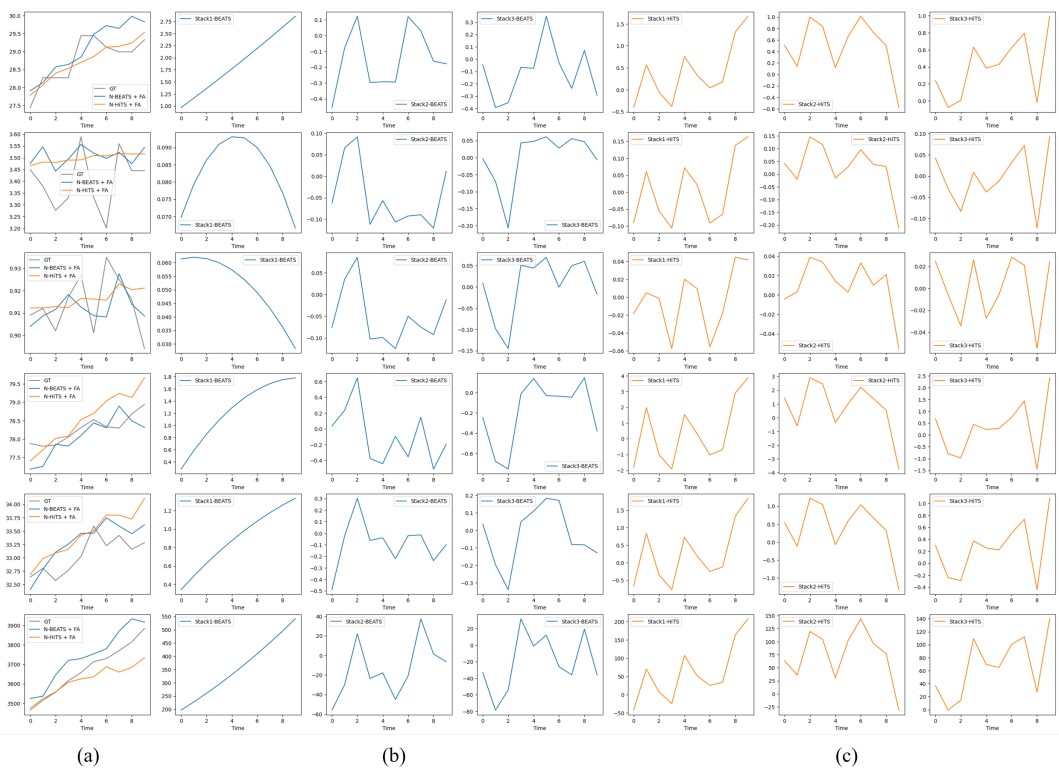

Figure 7: Visual analysis of interpretability. (a) Model forecasts, (b) stack forecasts of N-BEATS-I, and (c) N-HiTS. Note that N-BEATS-I utilizes a single trend stack and two seasonality stacks, sequentially.

**Visualization of representation.** We further investigate the representational landscape, we analyze the samples of pushforward measure from N-BEATS-I and N-HiTS. Adopting visualization techniques for both aligned and non-aligned instances as depicted in Figure 2, we configure UMAP with 5 neighbors, a minimum distance of 0.1, and employ the Euclidean metric. Similar to N-BEATS-G, we discern two observations in Figure 8 pertaining to N-BEATS-I and N-HiTS: (1) instances coalesce, residing closer to one another, and (2) an evident surge in domain entropy, from both N-BEATS-I and N-HiTS.

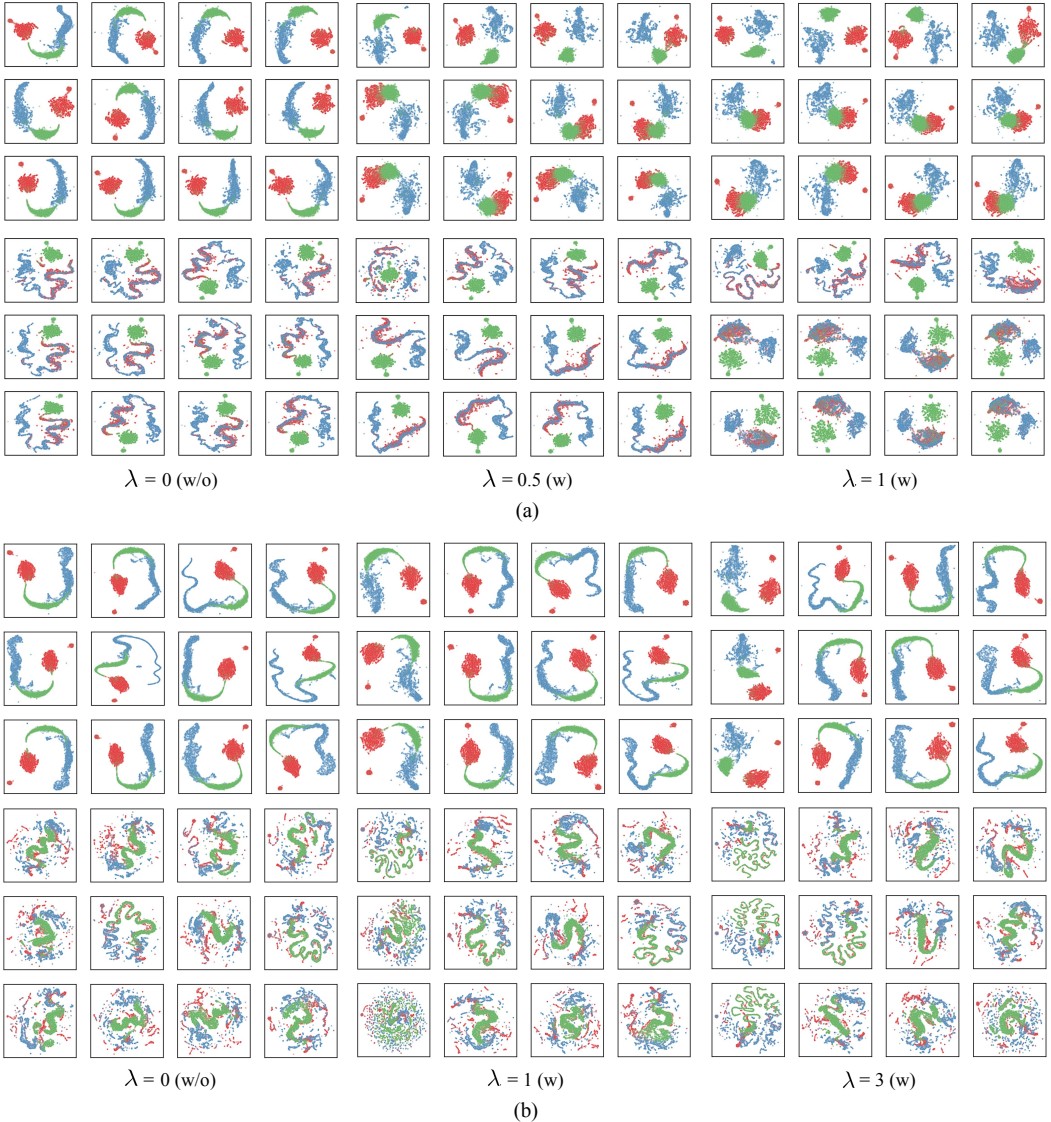

Figure 8: Visualization of extracted features. (a) N-BEATS-I, and (b) N-HiTS. For both (a) and (b), former plots illustrate increased inter-instance proximity, while subsequent ones depict inflated entropy.

# G  ABLATION STUDIES

## G.1  STACK-WISE VS BLOCK-WISE ALIGNMENTS

As mentioned in Remark 3.3, redundant gradient flows from recurrent architecture potentially causes gradient explosion or vanishing. To empirically validate this insight applied to our approach, we contrast stack-wise and block-wise feature alignments, as shown in Table 8. Notably, although stack-wise alignment generally outperform its counterpart, we do not observe the aforementioned problems, which could be identified by divergence of training. N-BEATS-I with block-wise alignment even demonstrates superior performance. Two plausible explanations are: (1) the limited number of stacks, and (2) the operational differences between the trend and seasonality modules in N-BEATS-I, which might help alleviating redundancy issue. Nonetheless, our primary objective of generalizing the recurrent model across various domains appears achievable through stack-wise alignment.

Table 8: Ablation study on alignment frequency (i.e., stack-wise vs block-wise alignments)

| Models | | N-HiTS | | | | N-BEATS-I | | | | N-BEATS-G | | | |
|---|---|---|---|---|---|---|---|---|---|---|---|---|---|
| Alignments | | Block-wise | | Stack-wise (Ours) | | Block-wise | | Stack-wise (Ours) | | Block-wise | | Stack-wise (Ours) | |
| Metrics | | sMAPE | MASE | sMAPE | MASE | sMAPE | MASE | sMAPE | MASE | sMAPE | MASE | sMAPE | MASE |
| **ODG** | | | | | | | | | | | | | |
| FRED | Commodity | 0.137 | 0.049 | **0.103** | **0.046** | 0.133 | 0.049 | 0.258 | 0.069 | 0.137 | 0.049 | 0.136 | 0.049 |
| | Income | 0.306 | 0.056 | **0.298** | **0.055** | 0.305 | **0.055** | 0.335 | 0.075 | 0.306 | 0.056 | 0.304 | **0.055** |
| | Interest rate | 0.121 | 0.074 | **0.100** | 0.070 | 0.119 | 0.073 | 0.189 | **0.046** | 0.121 | 0.075 | 0.120 | 0.073 |
| | Exchange rate | 0.040 | 0.060 | **0.034** | 0.058 | 0.040 | 0.060 | 0.075 | 0.069 | 0.040 | 0.060 | 0.039 | 0.060 |
| | Average | 0.151 | 0.060 | **0.134** | **0.057** | 0.149 | 0.059 | 0.214 | 0.065 | 0.151 | 0.060 | 0.150 | 0.059 |
| NCEI | Pressure | 0.368 | 0.255 | **0.350** | **0.250** | 0.367 | 0.254 | 0.409 | 0.305 | 0.368 | 0.255 | 0.367 | 0.255 |
| | Rain | 1.806 | 1.094 | 1.804 | **0.910** | 1.806 | 1.094 | **1.793** | 1.430 | 1.807 | 1.091 | 1.806 | 0.918 |
| | Temperature | 0.247 | 0.245 | **0.245** | **0.243** | 0.247 | 0.245 | 0.246 | 0.255 | 0.247 | 0.245 | 0.246 | 0.244 |
| | Wind | 0.456 | 0.645 | 0.454 | **0.644** | 0.456 | 0.645 | **0.449** | 0.662 | 0.457 | 0.645 | 0.454 | 0.645 |
| | Average | 0.719 | 0.560 | **0.713** | **0.512** | 0.719 | 0.560 | 0.724 | 0.663 | 0.720 | 0.559 | 0.718 | 0.516 |
| **CDG** | | | | | | | | | | | | | |
| FRED | Commodity | 0.102 | 0.046 | **0.081** | **0.044** | 0.105 | 0.046 | 0.197 | 0.059 | 0.108 | 0.047 | 0.107 | 0.046 |
| | Income | 0.301 | 0.054 | **0.295** | **0.053** | 0.301 | 0.055 | 0.323 | 0.070 | 0.301 | 0.054 | **0.295** | **0.053** |
| | Interest rate | 0.099 | 0.076 | **0.085** | 0.074 | 0.097 | 0.076 | 0.146 | **0.054** | 0.101 | 0.078 | 0.100 | 0.077 |
| | Exchange rate | 0.031 | 0.056 | **0.029** | **0.055** | 0.031 | 0.098 | 0.055 | 0.063 | 0.032 | **0.055** | 0.031 | **0.055** |
| | Average | 0.133 | 0.058 | **0.123** | **0.057** | 0.134 | 0.069 | 0.179 | 0.062 | 0.136 | 0.059 | 0.133 | 0.058 |
| NCEI | Pressure | 0.371 | 0.263 | 0.372 | **0.260** | **0.370** | 0.262 | 0.405 | 0.299 | 0.373 | 0.264 | 0.372 | 0.263 |
| | Rain | 1.809 | 1.144 | 1.803 | **0.783** | 1.808 | 1.148 | **1.787** | 1.831 | 1.804 | 1.181 | 1.804 | 1.178 |
| | Temperature | 0.246 | 0.244 | 0.245 | **0.242** | 0.246 | 0.244 | **0.240** | 0.244 | 0.246 | 0.244 | 0.245 | 0.243 |
| | Wind | 0.453 | 0.644 | 0.452 | **0.643** | 0.454 | 0.644 | **0.439** | 0.646 | 0.453 | **0.643** | 0.452 | **0.643** |
| | Average | 0.720 | 0.574 | **0.718** | **0.482** | 0.720 | 0.575 | **0.718** | 0.755 | 0.719 | 0.583 | **0.718** | 0.582 |
| **IDG** | | | | | | | | | | | | | |
| FRED | Commodity | 0.081 | 0.045 | **0.068** | **0.043** | 0.075 | 0.044 | 0.126 | 0.053 | 0.074 | 0.044 | 0.080 | 0.044 |
| | Income | 0.302 | 0.056 | **0.297** | **0.054** | 0.301 | 0.056 | 0.302 | 0.055 | 0.302 | 0.056 | 0.298 | 0.055 |
| | Interest rate | 0.079 | 0.083 | **0.071** | 0.081 | 0.079 | 0.082 | 0.086 | 0.091 | 0.079 | 0.083 | 0.074 | **0.081** |
| | Exchange rate | 0.025 | 0.051 | **0.024** | 0.050 | 0.025 | 0.051 | 0.029 | 0.056 | 0.025 | 0.051 | **0.024** | **0.050** |
| | Average | 0.122 | 0.059 | **0.115** | **0.057** | 0.120 | 0.058 | 0.136 | 0.064 | 0.120 | 0.059 | 0.119 | 0.058 |
| NCEI | Pressure | 0.384 | 0.276 | 0.384 | 0.272 | **0.383** | 0.277 | 0.389 | **0.263** | 0.384 | 0.276 | 0.384 | 0.276 |
| | Rain | 1.818 | 1.681 | **1.776** | **1.208** | 1.798 | 1.535 | 1.805 | 3.046 | 1.817 | 1.676 | **1.776** | **1.208** |
| | Temperature | 0.247 | 0.243 | 0.247 | 0.242 | 0.245 | 0.242 | **0.231** | **0.227** | 0.246 | 0.243 | 0.245 | 0.241 |
| | Wind | 0.453 | 0.641 | 0.452 | 0.640 | 0.453 | 0.641 | **0.434** | **0.623** | 0.453 | 0.641 | 0.452 | 0.640 |
| | Average | 0.726 | 0.710 | 0.715 | **0.591** | 0.720 | 0.674 | 0.715 | 1.039 | 0.725 | 0.709 | **0.714** | **0.591** |

## G.2 NORMALIZATION FUNCTIONS

According to the Table 9, Feature-aligned N-BEATS generally achieves superior performance when utilizing $\mathrm{softmax}$ function. However, there are instances where $\tanh$ function or even the absence of a normalization yields better results compared to the $\mathrm{softmax}$. This suggests that while scale is predominant instance-wise attribute, it may exhibit domain-dependent characteristics under certain conditions. Aligning this scale is therefore necessary. This entails that the $\mathrm{softmax}$, $\tanh$, and to not normalize offer different levels of flexibility in modulating or completely disregarding the scale information, implying a spectrum of capacities in aligning domain-specific attributes.

Table 9: Ablation study on normalization functions.

| Models | | N-HiTS | | | | | | N-BEATS-I | | | | | | N-BEATS-G | | | | | |
|---|---|---|---|---|---|---|---|---|---|---|---|---|---|---|---|---|---|---|---|
| Normalizers | | None | | tanh | | softmax (Ours) | | None | | tanh | | softmax (Ours) | | None | | tanh | | softmax (Ours) | |
| Metrics | | sMAPE | MASE | sMAPE | MASE | sMAPE | MASE | sMAPE | MASE | sMAPE | MASE | sMAPE | MASE | sMAPE | MASE | sMAPE | MASE | sMAPE | MASE |
| **ODG** | | | | | | | | | | | | | | | | | | | |
| FRED | Commodity | 0.103 | **0.046** | 0.104 | **0.046** | 0.103 | **0.046** | 0.265 | 0.070 | 0.270 | 0.069 | 0.258 | 0.069 | **0.050** | 0.136 | **0.050** | 0.135 | 0.136 | 0.049 |
| FRED | Income | 0.299 | 0.056 | 0.299 | 0.056 | **0.298** | **0.055** | 0.319 | 0.060 | 0.324 | 0.063 | 0.335 | 0.075 | 0.306 | 0.056 | 0.305 | 0.056 | 0.304 | **0.055** |
| FRED | Interest rate | 0.101 | 0.071 | 0.101 | 0.071 | **0.100** | 0.070 | 0.191 | 0.073 | 0.193 | 0.048 | 0.189 | **0.046** | 0.120 | 0.072 | 0.123 | 0.074 | 0.120 | 0.073 |
| FRED | Exchange rate | **0.034** | 0.058 | **0.034** | 0.058 | **0.034** | 0.058 | 0.072 | 0.061 | 0.077 | 0.074 | 0.075 | 0.069 | 0.041 | 0.061 | 0.043 | 0.059 | 0.039 | 0.060 |
| FRED | Average | 0.134 | 0.058 | 0.135 | 0.058 | 0.134 | **0.057** | 0.212 | 0.066 | 0.216 | 0.064 | 0.214 | 0.065 | **0.129** | 0.081 | 0.130 | 0.081 | 0.150 | 0.059 |
| NCEI | Pressure | 0.349 | 0.250 | **0.348** | 0.249 | 0.350 | 0.250 | 0.398 | 0.289 | 0.411 | 0.300 | 0.409 | 0.305 | 0.352 | **0.247** | 0.361 | 0.253 | 0.367 | 0.255 |
| NCEI | Rain | 1.819 | 0.918 | 1.820 | 0.917 | 1.804 | **0.910** | 1.808 | 2.087 | 1.807 | 1.841 | **1.793** | 1.430 | 1.814 | 1.075 | 1.814 | 1.071 | 1.806 | 0.918 |
| NCEI | Temperature | 0.247 | 0.244 | 0.246 | 0.244 | **0.245** | 0.243 | 0.248 | 0.253 | 0.249 | 0.256 | 0.246 | 0.255 | 0.247 | 0.245 | 0.248 | 0.245 | 0.246 | 0.244 |
| NCEI | Wind | 0.455 | 0.645 | 0.455 | **0.644** | 0.454 | **0.644** | 0.452 | 0.660 | 0.451 | 0.661 | **0.449** | 0.662 | 0.456 | 0.645 | 0.457 | 0.645 | 0.454 | 0.645 |
| NCEI | Average | 0.718 | 0.514 | 0.717 | 0.514 | **0.713** | **0.512** | 0.727 | 0.822 | 0.730 | 0.765 | 0.724 | 0.663 | 0.717 | 0.553 | 0.720 | 0.554 | 0.718 | 0.516 |
| **CDG** | | | | | | | | | | | | | | | | | | | |
| FRED | Commodity | 0.082 | 0.045 | **0.081** | **0.044** | **0.081** | **0.044** | 0.189 | 0.059 | 0.203 | 0.061 | 0.197 | 0.059 | 0.108 | 0.047 | 0.109 | 0.047 | 0.107 | 0.046 |
| FRED | Income | 0.296 | 0.055 | 0.296 | 0.055 | **0.295** | **0.053** | 0.323 | 0.088 | 0.319 | 0.064 | 0.323 | 0.070 | 0.302 | 0.054 | 0.301 | 0.054 | **0.295** | **0.053** |
| FRED | Interest rate | **0.085** | 0.074 | **0.085** | 0.075 | **0.085** | 0.074 | 0.145 | **0.052** | 0.149 | 0.058 | 0.146 | 0.054 | 0.101 | 0.078 | 0.101 | 0.077 | 0.100 | 0.077 |
| FRED | Exchange rate | **0.029** | 0.056 | **0.029** | 0.056 | **0.029** | 0.055 | 0.053 | 0.066 | 0.055 | 0.065 | 0.055 | 0.063 | 0.032 | 0.056 | 0.032 | 0.056 | 0.031 | **0.055** |
| FRED | Average | **0.123** | 0.058 | **0.123** | 0.058 | **0.123** | **0.057** | 0.178 | 0.066 | 0.182 | 0.062 | 0.179 | 0.062 | 0.136 | 0.059 | 0.136 | 0.059 | 0.133 | 0.058 |
| NCEI | Pressure | 0.373 | **0.260** | 0.374 | **0.260** | 0.372 | **0.260** | 0.405 | 0.316 | 0.410 | 0.313 | 0.405 | 0.299 | **0.255** | 0.372 | 0.257 | 0.356 | 0.372 | 0.263 |
| NCEI | Rain | 1.808 | 0.931 | 1.808 | 0.931 | 1.803 | **0.783** | 1.802 | 2.152 | 1.802 | 2.144 | **1.787** | 1.831 | 1.805 | 1.18 | 1.805 | 1.186 | 1.804 | 1.178 |
| NCEI | Temperature | 0.246 | 0.243 | 0.246 | 0.243 | 0.245 | 0.242 | 0.242 | 0.246 | 0.244 | 0.248 | **0.240** | 0.244 | 0.245 | 0.243 | 0.246 | 0.243 | 0.245 | 0.243 |
| NCEI | Wind | 0.453 | **0.643** | 0.453 | **0.643** | 0.453 | **0.643** | 0.442 | 0.649 | 0.443 | 0.649 | **0.439** | 0.646 | 0.452 | **0.643** | 0.453 | **0.643** | 0.452 | **0.643** |
| NCEI | Average | 0.720 | 0.519 | 0.720 | 0.519 | **0.718** | **0.482** | 0.723 | 0.841 | 0.725 | 0.839 | **0.718** | 0.755 | 0.737 | 0.562 | 0.738 | 0.560 | **0.718** | 0.582 |
| **IDG** | | | | | | | | | | | | | | | | | | | |
| FRED | Commodity | **0.068** | 0.044 | **0.068** | 0.044 | **0.068** | **0.043** | 0.124 | 0.052 | 0.142 | 0.055 | 0.126 | 0.053 | 0.083 | 0.045 | 0.083 | 0.045 | 0.080 | 0.044 |
| FRED | Income | 0.299 | 0.057 | 0.299 | 0.058 | **0.297** | **0.054** | 0.310 | 0.059 | 0.308 | 0.058 | 0.302 | 0.055 | 0.302 | 0.056 | 0.302 | 0.057 | 0.298 | 0.055 |
| FRED | Interest rate | 0.072 | 0.081 | 0.072 | **0.077** | 0.071 | 0.081 | 0.097 | 0.087 | 0.104 | 0.079 | 0.086 | 0.091 | 0.080 | 0.081 | 0.080 | 0.080 | 0.074 | 0.081 |
| FRED | Exchange rate | 0.025 | 0.051 | 0.025 | **0.050** | 0.024 | **0.050** | 0.033 | 0.055 | 0.040 | 0.055 | 0.029 | 0.056 | 0.026 | 0.052 | 0.026 | 0.051 | 0.024 | **0.050** |
| FRED | Average | 0.116 | 0.058 | 0.116 | **0.057** | 0.115 | **0.057** | 0.141 | 0.063 | 0.149 | 0.062 | 0.136 | 0.064 | 0.123 | 0.059 | 0.123 | 0.058 | 0.119 | 0.058 |
| NCEI | Pressure | 0.393 | 0.273 | 0.393 | 0.273 | 0.384 | 0.272 | 0.373 | **0.256** | 0.390 | 0.266 | 0.389 | 0.263 | **0.366** | 0.274 | 0.367 | 0.283 | 0.384 | 0.276 |
| NCEI | Rain | **1.776** | 1.211 | **1.776** | **1.205** | **1.776** | 1.208 | 1.883 | 3.222 | 1.873 | 3.848 | 1.805 | 3.046 | 1.818 | 1.671 | 1.818 | 1.695 | **1.776** | 1.208 |
| NCEI | Temperature | 0.247 | 0.242 | 0.247 | 0.242 | 0.247 | 0.242 | 0.236 | 0.232 | 0.235 | 0.231 | **0.231** | 0.227 | 0.246 | 0.241 | 0.246 | 0.242 | 0.245 | 0.241 |
| NCEI | Wind | 0.454 | 0.641 | 0.454 | 0.640 | 0.452 | 0.640 | 0.441 | 0.631 | 0.441 | 0.632 | **0.434** | 0.623 | 0.453 | 0.642 | 0.452 | 0.642 | 0.452 | 0.640 |
| NCEI | Average | 0.718 | 0.592 | 0.718 | **0.590** | 0.715 | 0.591 | 0.733 | 1.085 | 0.735 | 1.244 | 0.715 | 1.039 | 0.721 | 0.707 | 0.721 | 0.716 | **0.714** | 0.591 |

## G.3 SUBTLE DOMAIN SHIFT

Although the domain generalization commonly focuses on the domain shift problems, models may not perform as expected when the domain shift between source and target data is minimal. In some cases where the data from both domains align closely, fitting to source domain without invariant feature learning even can be beneficial. To examine this concern, we extend our analysis to the generalizability of Feature-aligned N-BEATS under such conditions. Table 10 demonstrates, while our model remains competitive, there is performance degradation observed in certain instances.

Table 10: Evaluation under subtle domain shift. 'F' and 'N' represent the FRED and NCEI datasets, respectively. The number of domains associated with each dataset is denoted accordingly, e.g., 'F3' represents three source domains from FRED. We conduct experiments by considering all possible combinations for each case.

| Methods | N-HiTS | | + FA (Ours) | | N-BEATS-I | | + FA (Ours) | | N-BEATS-G | | + FA (Ours) | |
|---|---|---|---|---|---|---|---|---|---|---|---|---|
| Metrics | sMAPE | MASE | sMAPE | MASE | sMAPE | MASE | sMAPE | MASE | sMAPE | MASE | sMAPE | MASE |
| F3 | **0.023** | 2.055 | **0.023** | 2.055 | 0.025 | **2.028** | 0.027 | 2.061 | 0.024 | 2.064 | 0.024 | 2.066 |
| F2N1 | 0.236 | 0.244 | 0.236 | 0.240 | 0.210 | 0.221 | **0.209** | **0.220** | 0.236 | 0.241 | 0.236 | 0.244 |
| F1N2 | 0.235 | 0.240 | 0.235 | 0.240 | **0.209** | 0.220 | **0.209** | **0.219** | 0.235 | 0.240 | 0.234 | 0.241 |
| N3 | 0.243 | 0.243 | 0.243 | 0.243 | **0.220** | **0.224** | 0.221 | 0.225 | 0.242 | 0.243 | 0.241 | 0.242 |
| Average | 0.184 | 0.695 | 0.184 | 0.694 | **0.166** | **0.673** | **0.166** | 0.680 | 0.184 | 0.697 | 0.184 | 0.698 |

## G.4 TOURISM, M3 AND M4 DATASETS

We extend our experimental scope to include three additional datasets: Tourism [3], M3 [36], and M4 [37]. Models are trained on two datasets and tested on the remaining dataset, enabling us to evaluate both ODG (M3, M4 → Tourism) and CDG (M3, Tourism → M4 and M4, Tourism → M3) scenarios. Our proposed methods consistently outperform N-BEATS models, demonstrating their generalization ability.

Table 11: Domain generalization performance on Tourism, M3 and M4 datasets. The first column indicates the target domain.

| Methods | N-HiTS | | + FA (Ours) | | N-BEATS-I | | + FA (Ours) | | N-BEATS-G | | + FA (Ours) | |
|---|---|---|---|---|---|---|---|---|---|---|---|---|
| Metrics | sMAPE | MASE | sMAPE | MASE | sMAPE | MASE | sMAPE | MASE | sMAPE | MASE | sMAPE | MASE |
| ODG | | | | | | | | | | | | |
| Tourism | 0.437 | 0.122 | 0.427 | 0.117 | 0.382 | 0.104 | **0.372** | **0.098** | 0.440 | 0.125 | 0.427 | 0.121 |
| CDG | | | | | | | | | | | | |
| M3 | 0.357 | 0.296 | 0.356 | 0.286 | 0.294 | 0.355 | **0.284** | 0.343 | 0.364 | 0.296 | 0.352 | **0.285** |
| M4 | 0.097 | 0.015 | 0.091 | **0.009** | 0.152 | 0.093 | 0.148 | 0.086 | 0.091 | 0.014 | **0.084** | **0.009** |

