# OpenReview forum: "Feature-aligned N-BEATS with Sinkhorn divergence"
_ICLR.cc/2024/Conference — ICLR 2024 spotlight_

### Official Review · Reviewer_g8xd · 2023-10-31

**Soundness:** 3 good
**Presentation:** 4 excellent
**Contribution:** 3 good
**Rating:** 8
**Confidence:** 3

**Summary:**

The paper introduces a new algorithm for performing time series forecasting under domain shift. It does so by combining the N-BEATS deep learning model for time series forecasting with a feature alignment regularization that is intended to select for robust models across a variety of domains. While both approaches had been previously introduced and popularized in separate bodies of literature, the combination of the two approaches is nontrivial due to the challenges of implementing feature alignment on a stack-like architecture.

Concretely, the authors parameterize the N-BEATS architecture with a collection of feature mappings $\phi^m$, forecast generators $\xi_{\downarrow}$, and backcast generators $\xi_{\uparrow}$. Each stack produces a prediction of the forecast $\hat{y}$ by repeatedly computing residual features and mapping those to forecasts and backcasts, whose residuals are used to generate further features. The predictions of multiple stacks are combined together. The paper introduces a feature alignment regularization term, which is small when the  features extracted by each stack are distributionally similar across domains. While the $\mathcal{H}$-divergence initially seems like the proper quantity to measure distributional distance, the quantity is extremely computationally expensive to estimate. Instead, the authors use think entropically-regularized Sinkhorn divergence to measure the difference between extracted feature distributions. They package these insights into an training algorithm in Section 3.3 that alternately updates the parameters of different component networks.

The authors provide some theoretical analysis of the algorithm's properties, without including no new optimization or generalization bounds. Lemma 3.4 bounds the Lipschitz smoothness of each stack's feature extraction map, while Theorem 3.6 bounds the extracted features' maximum of the Sinkhorn divergences by the maximum of the divergence between domain input distributions, which depends on the aforementioned Lipschitz bound.

They empirically evaluate the performance of the algorithm in Section 4 by comparing the performance of these algorithms with alternatives on time series datasets like FRED and NCEI. They show marked improvements in outside domain generalization over N-BEATS algorithms without feature alignment and post comparable performance among in-domain generalization. They also compare several levels of entropic regularization and demonstrate a sharp improvement in runtime over computing Wasserstein-2 divergences, while maintaining similar performance.

**Strengths:**

As far as the reviewer (who is not an expert in either domain shift or time series forecasting) is aware, the paper is a novel and interesting contribution to both fields with compelling empirical results and strong application of optimal transport methods. The crafting of the feature alignment regularization term appears to be original for time series forecasting, as is the application of Sinkhorn divergence. Given the challenges of applying neural networks to time series data and the particularly pressing domain shift challenges posed there, the problem seems relevant, and the paper's introduction does a good job making that clear.

The paper is easy to read and includes clear explanations of the purposes of each component of the algorithm. Figure 2 is a particularly helpful illustration of the neural network architecture. Notation is clear overall with few mistakes, and the reviewer detected no major mathematical errors.

The experiments are thorough and compare the method with a wide range of alternatives, considering both runtime and accuracies.

**Weaknesses:**

While I am overall satisfied with the contributions of the papers, there are several issues that I would like to see addressed by the authors.

For ease of understanding, I think that either the Contributions paragraph or a different subsection before Section 2 could be used to more clearly signpost how the algorithm will be presented in Sections 2 and 3. Including something like equation (3.7) earlier to demonstrate the kind of training objective being constructed could be helpful for readers to identify the role played by feature alignment throughout. A large number of parameters are introduced in these sections, such as $\alpha$, $\beta$, $\gamma$, $K$, $M$, and $L$; I think some of these could be better contextualized, so that the reader can know which are expected to be large and which small. (For instance, we should probably think of $L$ and $M$ as relatively small constants, due to their impact on Lipschitzness.)

While Table 2 of the empirical results compares the runtime of computing different divergences, I'd be interested in seeing comparisons between the end-to-end runtimes of the methods in Table 1, in order to evaluate whether the computing the regularization term for feature alignment adds a substantial computational cost.

As the authors state, the theoretical contributions of the paper are relatively light and focus on bounding the Sinkhorn divergence of extracted features. Future proofs of convergence (or experimental analyses thereof) and generalization bounds for Sinkhorn divergence in the vein of Proposition 2.1 would be valuable, but I do not consider these essential inclusions in this work for publication.

# Minor comments
- The clause on pg 1 "the proposed model is built a purely deep learning-based model which is N-BEATS proposed by [42,43]" parses a little awkwardly.
- $\epsilon$ is overloaded as both the entropic regularization parameter and the expected risk in Proposition 2.1, which is never explicitly defined.
- In Definition 3.1, should $m=1, \dots, d$ instead end with $M$?
- The forecasting algorithms N-BEATS-I and N-BEATS-G are seen for the first time in Section 4 without context; a pointer to the first page of the appendix---where they are helpfully disambiguated---would be appreciated.
- Given the number of entries and small fonts in the tables (like Table 1), bolding the values of the best losses in each row may improve readability.

**Questions:**

Are the empirical time series tasks and the formulations of ODG, CDG, and IDG standard and comparable to other works? I noticed that the original N-BEATS paper was evaluated on M3 and M4 competition datasets, as well as a Tourism dataset. To the authors and other reviewers, are there other standard datasets that should be considered for this method to be empirically compelling?

Is there any intuition or experimental results that comment on the smoothness of $\sigma \circ g^m$? While the Lipschitzness bound in Lemma 3.4 suggests it could be highly non-smooth, I'd be interested to know more about the gradients were largely well-behaved.

What should I take away from Figure 1? It appears that the structure of the embeddings changes little between stacks and blocks, and it's unclear to me whether that should be the case or not. The structural differences between $\lambda = 1$ and $\lambda = 3$ are almost undetectable.

---

> ### Author Response · Authors · 2023-11-19
>
> # Response to Weakness 1
>
> We are grateful for the insightful suggestions, especially those aimed at improving our presentation. For ease of understanding, before starting the main section (Sections 2 and 3), we have signposted how the training objective with the corresponding algorithm will be presented,
>
> > The remainder of the article is organized as follows. In Section 2, we set the domain generalization problem in the context of time series forecasting, review the doubly residual stacking architecture of N-BEATS, and introduce the error analysis for domain generalization models. Section 3 is devoted to defining the marginal feature measures inspiring the stack-wise alignment, introducing the Sinkhorn divergence together with the corresponding representation learning bound, and presenting the training objective with the corresponding algorithm. In particular, Figure 1 therein illustrates the overall architecture of Feature-aligned N-BEATS. In Section 4, comprehensive experimental evaluations are provided. Section 5 concludes the paper. Other technical descriptions, visualized results, and ablation studies are given in Appendix.
>
> In addition, we have contextualized $\alpha$, $\beta$, $\gamma$, $K$, $M$, and $L$ in 'Experimental details' in Section 4.
>
> > The lookback horizon, forecast horizon, and the number of source domains are set to be $\alpha=50$, $\beta=10$, and $K=3$, respectively (noting that it depends on the characteristics of source domains' datasets). Furthermore, the number of stacks and blocks, and the dimension of feature space are set to be $M=3$, $L=4$, and $\gamma=512$, respectively (noting that it is consistent with N-BEATS [45]).
>
> # Response to Weakness 2
>
> We appreciate the suggestion for evaluating the potential increase in computational cost due to the introduction of FA.
> We have added a new row to Table 1, which presents the training time per iteration.
>
> |               | N-HiTS | + FA (Ours) | N-BEATS-I | + FA (Ours) | N-BEATS-G | + FA (Ours) | NLinear | DLinear | Autoformer | Informer |
> | ------------- | ------ | ----------- | --------- | ----------- | --------- | ----------- | ------- | ------- | ---------- | -------- |
> | Runtime (sec) | 0.26   | 0.80        | 0.32      | 0.97        | 0.16      | 0.68        | 0.04    | 0.05    | 0.58       | 0.50     |
>
> The updated table reveals that stack-wise alignment does increase the training time compared to N-BEATS.
> However, Table 2 demonstrates that SD-based alignment, the chosen regularizer, exhibits the most enhanced generalizability while incurring marginal computational cost.
> Please note that there is no difference in runtime between Feature-aligned N-BEATS and original N-BEATS during inference, since alignment is not applied during this stage.
>
> # Response to Weakness 3
>
> We greatly appreciate the suggestion for future work and concur with its significance. We earnestly hope that our empirical findings regarding the potential of feature alignment adaptation in time series forecasting models will contribute valuable insights for future research.
>
> # Response to Comment 1
>
> Thanks for this careful advice. We have revised this sentence to 'In particular, the proposed model is built upon a deep learning model which is N-BEATS [45, 46]'.
>
> # Response to Comment 2
>
> Thanks for point out our mistake. We have replaced the notation for the expected risk to $R^T, R^k$. Furthermore, we denoted the notations before Proposition 2.1.
>
> > Furthermore, denote by $R^k(\cdot):{\cal H}\rightarrow \mathbb{R}$ and $R^T(\cdot):{\cal H}\rightarrow \mathbb{R}$ the expected risk under the source measures $\mathbb{P}^k$, $k=1,\dots,K$, and the target measure $\mathbb{P}^T$, respectively.
>
> # Response to Comment 3
>
> Sorry for this mistake. We have revised it. Please refer to Definition 3.1 in Section 3.1.
>
> # Response to Comment 4
>
> Following this suggestion, we have added the pointer in the caption of Table 1.
>
> # Response to Comment 5
>
> To improve readability, we have bolded the best performance in each row of the all tables.
> We appreciate the suggestion, which significantly improves the clarity of our paper.

---

> ### Author Response · Authors · 2023-11-19
>
> # Response to Question 1
>
> In this paper, we tackle the domain generalization (DG) scenario in which multiple source domains (i.e., $K \geq 2$) are utilized to make predictions for a distinct target domain. This contrasts with single domain generalization (SDG), which involves only one source domain. In this context, N-BEATS as presented in [46] lies on the SDG paradigm from a meta-learning standpoint. Hence, their experimental setup is not directly transferable to our DG context.
> Nevertheless, we have adapted the datasets from the N-BEATS study and its corresponding results have been added in Appendix G.4 and Table 11.
> The results demonstrate that Feature-aligned N-BEATS consistently outperform the original N-BEATS.
>
> |          |       | N-HiTS | + FA (Ours) | N-BEATS-I | + FA (Ours) | N-BEATS-G | + FA (Ours) |
> | -------- | ----- | ------ | ----------- | --------- | ----------- | --------- | ----------- |
> | M3, M4   | sMAPE | 0.437  | 0.427       | 0.382     | 0.372       | 0.440     | 0.427       |
> | -> Tour  | MASE  | 0.122  | 0.117       | 0.104     | 0.098       | 0.125     | 0.121       |
> | M4, Tour | sMAPE | 0.357  | 0.356       | 0.294     | 0.284       | 0.364     | 0.352       |
> | -> M3    | MASE  | 0.296  | 0.286       | 0.355     | 0.343       | 0.296     | 0.285       |
> | M3, Tour | sMAPE | 0.097  | 0.091       | 0.152     | 0.148       | 0.091     | 0.084       |
> | -> M4    | MASE  | 0.015  | 0.009       | 0.093     | 0.086       | 0.014     | 0.009       |
>
> # Response to Question 2
>
> Thanks for pointing this out.
> According to your suggestion, we have plotted the forecasting and alignment losses during both training and validation steps in Figure 5 in Appendix D.
> This illustrates stable training process, indicating that the gradients are well-behaved during training.
>
> > Feature-aligned N-BEATS is a complicated architecture based on the doubly residual stacking principle with an additional feature-aligning procedure. To investigate the stability of training, we analyze the training and validation loss plots. Figure 5 indicates that the gradients are well-behaved during training. This stable optimization is regarded to the Lemma 3.4 presented in Section 3.1.
>
> # Response to Question 3
>
> In Figure 1 (which is Figure 2 in revised version), we demonstrate 1) the effects of FA on the embedding space and 2) its sensitivity to the hyperparameter $\lambda$, with a focus on the alignment aspect (We typically used $\lambda$ values of 1 and 3 for N-BEATS-G due to their superior performance in varied experimental settings.). N-BEATS operates as a recurrent framework, with blocks in stacks capturing distinct temporal features in time series data, such as trends and seasonality. Our goal was to assess whether FA successfully narrows the gap between source domains concerning these diverse features. This involved analyzing all embedding spaces from every block across all stacks. In this regard, it becomes evident that the domains are closely aligned for most features. Although the difference between $\lambda = 1$ and $\lambda = 3$ appears subtle, indicating that N-BEATS-G with FA has low sensitivity to $\lambda$ variations}, we observed that $\lambda$ can be a critical factor in specific cases, such as with N-BEATS-I, as detailed in Appendix F (Figure 8).

---

> > ### Comment · Reviewer_g8xd · 2023-11-23
> >
> > I thank the authors for their detailed response and willingness to address my concerns, in particular the additional experiments done for Weakness 2 and and Question 1. I maintain my score.

---

> > > ### Author Response · Authors · 2023-11-23
> > >
> > > We greatly appreciate your positive evaluation on our paper and constructive comments which have further improved our paper.

---

### Official Review · Reviewer_adrw · 2023-10-31

**Soundness:** 3 good
**Presentation:** 2 fair
**Contribution:** 3 good
**Rating:** 6
**Confidence:** 3

**Summary:**

The paper proposes the advancement of N-BEATs, a pure deep learning based method for time series forecasting. The novelty of the approach is a combination of stack-wise feature alignment (FA) and N-BEATs. The training objective is to minimize the empirical risk and the stack-wise Sinkhorn divergences (Section 3.3) as an approximation of optimal transport measures for feature alignment.

Well written introduction points out that it is not a straightforward combination of the established components but a nontrivial extension of NBEATs that poses several challenges. The paper lays down background on NBEATs and FA in rather technical manner. Never the less it stresses the main challenges nicely and the way presented approach mitigates them, including proofs of the main theoretical steps, Lemma 3.4, Theorem 3.6 (Sinkhorn div.), eventually resulting in the Algorithm 1. The experimental section test the model on two real world datasets (FRED, NCEI) against three domain shift scenarios:  out-domain generalization (ODG), cross-domain generalization (CDG), and in-domain generalization (IDG) with on-par or superior (confidence intervals are not present) results.

**Strengths:**

+ A novel practical method with available code supported by theoretical results
+ Timely taps into lively research field and improves on previous existing results on pure deep learning based time series forecasting interesting to ML community

**Weaknesses:**

- Overall presentation: The main contributions are rather buried in the technical exposition of sections 1-3. The main idea (adding stack-wise FA through minimising Sinkhorn div. added to the N-BEATS loss) could be simply laid out and perhaps visualised while some of the technical details may be put into Appendices. I would consider starting from algorithm and Section 3.3. and than relate it to principles in a backward manner as opposed to a current forward exposition that simply takes too long and has several leaps of faith anyways (see below). Authors are encouraged to lighten the exposition up in the main paper body and increase the rigour int he Appendix. More over, if paper claims “extraordinary results” bringing some results (e.g.,  Fig.4 in Appendix or simplified Fig.1) forward into the Introduction could increase motivation of the readers.

- Due to multitude of approximation and heuristics used, i.e., Sinkhorn instead of H-divergence, Definition 3.5, Remark 3.8,  “... Instead, we conduct the alignment by regularizing exclusively on feature extractors $\Psi$ ...” the work effectively strongly relies on experimental results. Experimental results seems only marginally better compared to selected alternatives (Tables 1 and 2), however. These should be extended to increase the validity of the results.

- I believe important but unclear computational cost increase compared to alternatives (cost increase is only assumed by me but expected since method adds “FA” and more on top)

- One of the major features that N-BEATS algorithm provides is an Interpretability. The paper avoids any analysis of interpretability after adding FA

**Questions:**

1. Authors proposed stack-wise feature alignment using Sinkhorn divergence (original feature alignment uses KL divergence or Jensen-Shannon divergence). Can authors elaborate on the difference?

2. Would naive feature alignment applied on N-BEATS work? Why not? That is, why should reader consider using proposed method?

3. Paper already suggests using approximation to H-divergence which is computational expensive. So even more importantly, what are computation expenses related to proposed stack-wise approach compared to N-BEATS?

4. Results are only showing visually rather marginal gains (see Weaknesses) against the compared methods with differences on 2nd or even third decimal places (I know, it's percentage mean error, but still ...). Could authors elaborate more on the positive side of the results and put into context increased computational costs? In general I would suggest to discuss limitations and practical usefulness of the method.

Some further suggestions:
- Consider bringing, possibly simplified, model architecture (Fig 2 from Appendix) to the main body instead of a rather cumbersome technical description Section 2
- Focus more on novelty of the proposed approach in Sections 1-3.

---

> ### Author Response · Authors · 2023-11-19
>
> # Response to Weakness 1 and Suggestion 1
>
> We appreciate the valuable suggestions for our presentation. In response, we have provided a simplified illustration of Feature-aligned N-BEATS in Figure 1 in Section 3 of the revised script for better accessibility (noting that its detailed version is provided in Figure 3 in Appendix A). Additionally, we have relocated some of the more detailed technical aspects to the appendix for improved readability. Furthermore, for ease of understanding, before starting the main section (Sections 2 and 3), we have signposted how the training objective with the corresponding algorithm will be presented,
>
> > The remainder of the article is organized as follows. In Section 2, we set the domain generalization problem in the context of time series forecasting, review the doubly residual stacking architecture of N-BEATS, and introduce the error analysis for domain generalization models. Section 3 is devoted to defining the marginal feature measures inspiring the stack-wise alignment, introducing the Sinkhorn divergence together with the corresponding representation learning bound, and presenting the training objective with the corresponding algorithm. In particular, Figure 1 therein illustrates the overall architecture of Feature-aligned N-BEATS. In Section 4, comprehensive experimental evaluations are provided. Section 5 concludes the paper. Other technical descriptions, visualized results, and ablation studies are given in Appendix.
>
> We tried to lighten the exposition up in the main paper by putting some remarks (Remarks C.1 and C.2 in the revised script, which were Remarks 3.7 and 3.8 in the original script) in Appendix C.
>
> # Response to Weakness 2
>
> We kindly ask to take a careful look at the case of CDG and ODG with MASE metric in Table 1. Since CDG represents the case where the source domain data are distinctive from one another, we can say that the representation learning through the alignment of distinctive source domains' feature measures can help to improve the original model's forecasting power. Furthermore, since the ODG represents that the source data and target data belong to the different super domains (that is, the source and target data are distinctive from each other), we can say that our FA-NBEATS improves the forecasting power on the (unknown) distinctive target sequential data (when domain shift is significant). For the case of IDG, as the source data and target data belong to the same super domain (that is, the source and target data are relatively not distinctive from each other), the improvement seems to be marginal. But this is quite intuitive because the model would not have much chance to learn domain invariant features through the similar types of the source domain and the model's prediction power at the not distinctive target domain could not be improved through the feature alignment of similar distributions. In this regard, we further emphasize that our experiment performances are verified in diverse domain generalization environments, and the performances' improvement also stands out.
>
> Furthermore, our choice of SD over other divergences is substantiated by Table 2 in Section 4 and Table 6 in Appendix E for the following reasons: 1) While SD yields performance comparable to WD (noting that SD is an approximate of WD.), SD is more computationally efficient. 2) SD outperforms both MMD and KL (in the ODG context) in terms of forecasting, with MMD incurring higher computational costs compared to SD. 3) Despite KL being the most computationally efficient, Table 6 reveals its lack of robustness, as indicated by its frequent inability to produce a result, denoted as 'NaN'.
>
> # Response to Weakness 3 and Questions 3-4
>
> We appreciate the suggestion for evaluating the potential increase in computational cost due to the introduction of FA.
> We have added a new row to Table 1, which presents the training time per iteration.
>
> |               | N-HiTS | + FA (Ours) | N-BEATS-I | + FA (Ours) | N-BEATS-G | + FA (Ours) | NLinear | DLinear | Autoformer | Informer |
> | ------------- | ------ | ----------- | --------- | ----------- | --------- | ----------- | ------- | ------- | ---------- | -------- |
> | Runtime (sec) | 0.26   | 0.80        | 0.32      | 0.97        | 0.16      | 0.68        | 0.04    | 0.05    | 0.58       | 0.50     |
>
> The updated table reveals that stack-wise alignment does increase the training time compared to N-BEATS.
> However, Table 2 demonstrates that SD-based alignment, the chosen regularizer, exhibits the most enhanced generalizability while incurring marginal computational cost.
> Please note that there is no difference in runtime between Feature-aligned N-BEATS and original N-BEATS during inference, since alignment is not applied during this stage.

---

> ### Author Response · Authors · 2023-11-19
>
> # Response to Weakness 4
>
> We appreciate your concern about the interpretability of our model.
> In this revision, we have provided a visualization (Figure 7) and additional explanation in Appendix F for interpretability analysis,
>
> > Figure 7 exhibits the interpretability of the proposed method by presenting the final output of the model and intermediate stack forecasts. N-BEATS-I and N-HiTS presented in Appendix A have interpretability. More specifically, N-BEATS-I explicitly captures trend and seasonality information using polynomial and harmonic basis functions, respectively. N-HiTS employs Fourier decomposition and utilizes its stacks for hierarchical forecasting based on frequencies. Preserving these core architectures during the alignment procedure, Feature-aligned N-BEATS still retains interpretability.
>
> # Response to Question 1
>
> We consent on this advice. Conceptually, divergences such as Wasserstein Divergence (WD) [28], Sinkhorn Divergence (SD) [44], Maximum Mean Discrepancy (MMD) [30], and Kullback–Leibler (KL) Divergence [65] can be applied for the alignment loss. Our choice is supported by the experimental advantages of SD, which are 1) computational cost and 2) robustness on performance. It is empirically demonstrated in Table 2 in the main script and Table 6 in Appendix E. On top of that, SD enables us to leverage the theoretical properties of optimal transport problems (Lemma 3.4 and Theorem 3.6).
>
> # Response to Question 2
>
> We highlight that under doubly residual stacking architecture, defining feature alignment is nontrivial, which is well-described in the 4-th paragraph of the Introduction in the revised script.
>
> > First, N-BEATS does not allow the feature alignment in a 'one-shot' unlike the aforementioned references. This is because it is a hierarchical multi-stacking architecture in which each stack consists of several blocks and is connected to each other by residual operations and feature extractions.
>
> In this regard, we adopted a 'stack-wise' feature alignment strategy rather than 'naive' or 'block-wise' alignment. In particular, we believe that stack-wise alignment ensures wide feature coverage in the context of alignment and promotes efficient gradient descent. The N-BEATS framework, which employs a doubly residual stacking architecture, encodes diverse features across its blocks and stacks [45]. Our goal was to align these features by frequently calculating divergence (either block-wise or stack-wise) and incorporating this into the alignment loss. However, as [47] noted, overly frequent loss propagation can lead to issues of gradient vanishing or exploding. Consequently, we determined that 'stack-wise' alignment offers an optimal frequency for this process. A comparative analysis of block-wise and stack-wise alignment is detailed in Table 8 in Appendix G.1.
>
> # Response to Suggestion 2
>
> We already clarified our contributions explicitly in the 7-th paragraph of the Introduction (Section 1),
>
> > To provide an informative procedure of stack-wise feature alignment, we introduce a concrete mathematical formulation of N-BEATS (Section 2), which enables to define the pushforward feature measures induced by the intricate residual operations and the feature extractions for each stack (Section 3.1). From this, we make use of theoretical properties of optimal transport problems to show a representation learning bound for the stack-wise feature alignment with the Sinkhorn divergence (Theorem 3.6), which justifies the feasibility of Feature-aligned N-BEATS. To embrace comprehensive domain generalization scenarios, we use real-world data and evaluate the proposed method under three distinct protocols based on the domain shift degree. We show that the model consistently outperforms other forecasting models. In particular, our method exhibits outstanding generalization capabilities under severe domain shift cases (Table 1). We further conduct ablation studies to support the choice of the Sinkhorn divergence in our model (Table 2).

---

> ### Comment · Reviewer_adrw · 2023-11-23
>
> I thank to authors for their responses and addressing raised concerns. Especially a clarifying response to W2, pointing out FA addresses out-of-domain generalization well in experiments, and adding runtimes to Table 1, brought more confidence in the practical usefulness of the proposed method.
>
> Altogether, after having read remaining responses and revisions suggested, I increase my rating of the paper to 6 (weak accept). Thanks again for your contributions and good luck.

---

> > ### Author Response · Authors · 2023-11-23
> >
> > We greatly appreciate your reconsideration on our paper and valuable comments which have improved our paper.

---

### Official Review · Reviewer_F4wY · 2023-11-08

**Soundness:** 3 good
**Presentation:** 3 good
**Contribution:** 2 fair
**Rating:** 6
**Confidence:** 3

**Summary:**

* The paper proposes a new time series model "Feature-aligned N-BEATS", which is an extension of existing N-BEATS model targeting domain-generalized time series and learning invariant features.
* The novelty of this model is the introduction of alignment across different stacks using Sinkhorn divergence. The training loss includes both empirical risk minimization from multiple domains and an alignment loss calculated with Sinkhorn divergence to promote invariance.
* The paper also includes experiments and ablation studies, showing the model's effective forecasting and generalization capabilities​.

**Strengths:**

* The paper is well-written and easy to follow.
* The proposed idea is simple but yet effective given the experimental results.
* The distance measure choice of Sinkhorn divergence seems promising. It shows non-trivial improvements in the ablation study and also takes much less time compared with standard Wasserstein distance.

**Weaknesses:**

I checked the proposed architecture and the experimental results. They look convincing.

My concern is that the idea is not that "new". Imposing penalties on divergence is commonly used to promote model invariances. The novelty might be the application and the extension on N-BEATS model at the stacking level. I am not sure whether this contribution is enough. Another concern is that the analysis (theorem 3.6) seems to be direct extensions of the literature [41] which may diminish the contribution as well.

In general, I still believe it's a good paper and the proposed method seems working and elegant. However, my review might be superficial given I am not familiar with this domain.

**Questions:**

Please see above section.

---

> ### Author Response · Authors · 2023-11-19
>
> # Response to Weakness
>
> We sincerely appreciate this valuable advice.
> To underscore our specific contributions and further illustrate this point, we would like to emphasize the following:
>
> - Under doubly residual stacking architecture, defining feature alignment is nontrivial, which is well-described in the 4-th paragraph of the Introduction in the revised script.
>
>   > First, N-BEATS does not allow the feature alignment in a ‘one-shot’ unlike the aforementioned references. This is because it is a hierarchical multi-stacking architecture in which each stack consists of several blocks and is connected to each other by residual operations and feature extractions.
>
>   In this regard, we have put lots of effort into representing N-BEATS in a mathematical manner (noting that it does not appear in the original N-BEATS paper) in Section 2.2. This mathematical definition has allowed us to establish important theoretical foundations for learning invariance.
>
>   Based on this formal representation, we could achieve Lipschitz continuity of marginal feature maps (Lemma 3.4). Hence, by leveraging this with theoretical properties of optimal transport problems, we could establish representation learning bounds (Theorem 3.6). Therefore, it is not a direct extension of [41] where only simple fully-connected layers are considered.
> - We have carefully considered the optimal alignment frequency (Remark 3.3), resulting in the 'stack-wise' approach. This application is not simple but rather a rational design (see Figure 1 in the revised script), addressing the gradient flow challenges in recurrent models as discussed by [47] and leveraging the expressiveness characteristic of N-BEATS, as detailed by [45].
> - We have found an effective divergence for stack-wise feature alignment by extensive qualitative analysis with respect to computational cost and robustness performance (Table 2); that is, Sinkhorn divergence. To demonstrate the generalizability and forecasting power of our approach, we have presented results across a wide range of scenarios, including IDG, CDG, and ODG, all with real-world data (Evaluation details, Section 4). These protocols represent novel contributions to the field of forecasting (Table 1).

---

### Author Response · Authors · 2023-11-19

# Upload of Revised Script

We have attached the latest version of the script, in which modifications are highlighted with 'blue' color.

---

### Meta-Review · Area_Chair_YQMe · 2023-12-26

**Metareview:**

This paper proposes an extension of the N-BEATS approach used for univariate time series prediction. This model, "Feature-aligned N-BEATS" introduces an alignment across different stacks using the Sinkhorn divergence (SD), which is added to the regular loss (Eq. 3.7). The paper proposes an interesting ablation, studying various other metrics, showing that SD is superior to other possible choices, both for performance and speed. The 3 reviewers have greatly appreciated this paper and the authors have provided significant clarifications during the rebuttal.

**Justification For Why Not Higher Score:**

The paper could be selected as spotlight or poster, and is borderline in that sense. The work does not need to be put highlighted as oral.

**Justification For Why Not Lower Score:**

On the plus side, I find this is an interesting of the Sinkhorn algorithm, and performance improvements on 1D problem do matter for certain practitioners (forecasting). Experiments in domain generalization are convincing and show consistent improvements.

---

### Decision · Program_Chairs · 2024-01-16

Accept (spotlight)